# ELIAS: End-to-End Learning to Index and Search in Large Output Spaces

**Nilesh Gupta**
UT Austin
nilesh@cs.utexas.edu

**Patrick H. Chen**
UCLA
patrickchen@g.ucla.edu

**Hsiang-Fu, Yu** *
Amazon
rofu.yu@gmail.com

**Cho-Jui, Hsieh**
UCLA
chohsieh@cs.ucla.edu

**Inderjit S. Dhillon**
UT Austin & Google
inderjit@cs.utexas.edu

## Abstract

Extreme multi-label classification (XMC) is a popular framework for solving many real-world problems that require accurate prediction from a very large number of potential output choices. A popular approach for dealing with the large label space is to arrange the labels into a shallow tree-based index and then learn an ML model to efficiently search this index via beam search. Existing methods initialize the tree index by clustering the label space into a few mutually exclusive clusters based on pre-defined features and keep it fixed throughout the training procedure. This approach results in a sub-optimal indexing structure over the label space and limits the search performance to the quality of choices made during the initialization of the index. In this paper, we propose a novel method ELIAS which relaxes the tree-based index to a specialized weighted graph-based index which is learned end-to-end with the final task objective. More specifically, ELIAS models the discrete cluster-to-label assignments in the existing tree-based index as soft learnable parameters that are learned jointly with the rest of the ML model. ELIAS achieves state-of-the-art performance on several large-scale extreme classification benchmarks with millions of labels. In particular, ELIAS can be up to 2.5% better at precision@1 and up to 4% better at recall@100 than existing XMC methods. A PyTorch implementation of ELIAS along with other resources is available at https://github.com/nilesh2797/ELIAS.

## 1 Introduction

Many real-world problems require making accurate predictions from a large number of potential output choices. For example, search advertising aims to find the most relevant ads to a given search query from a large corpus of ads [26, 14], open-domain question answering requires finding the right answers to a given question from a large collection of text documents [8, 29], and product recommendation requires recommending similar or related products from a large product catalog, based on past searches and interactions by users. eXtreme Multi-label Classification (XMC) is a popular framework for solving such problems [4], which formulates these problems as a multi-label classification task with very large number of labels; here each output choice is treated as a separate label. A label $\ell$ is often parameterized by its *one-versus-all* classifier vector $\mathbf{w}_\ell$ and the relevance between label $\ell$ and input $\mathbf{x}$ is formulated as $\mathbf{w}_\ell^T \phi(\mathbf{x})$, where $\phi$ is an encoding function which maps an input $\mathbf{x}$ to its vector representation.

---

*This work does not relate to Hsiang-Fu's position at Amazon

36th Conference on Neural Information Processing Systems (NeurIPS 2022).

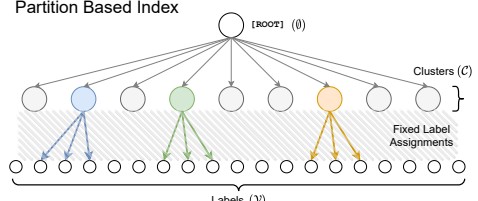 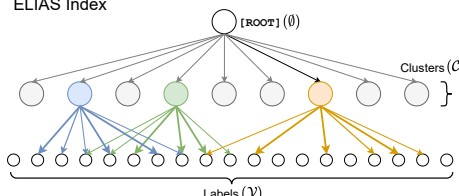

Figure 1: Traditional partition-based index vs ELIAS index; here an arrow from a cluster to a label denotes the assignment of the label to the cluster, arrow width indicates the weight of the assignment. (*left*) Existing partition based XMC methods use a shallow balanced tree as the index structure with a label uniquely assigned to exactly one cluster; moreover, they initialize the clusters over pre-defined features and keep them fixed throughout the training procedure. (*right*) ELIAS generalizes the tree based index to a sparsely connected graph-based index and learns the cluster-to-label assignments end-to-end with the task objective during training.

Evaluating $\mathbf{w}_\ell^T \phi(\mathbf{x})$ for every label $\ell$ in an XMC task can get computationally expensive since the number of labels could easily be upwards of millions. To reduce the complexity, most existing methods employ a search index that efficiently shortlists a small number of labels for an input query and the relevance scores are only evaluated on these shortlisted labels. The quality of the search index plays a pivotal role in the accuracy of these methods since a label $\ell$ outside the shortlist will be directly discarded, even if it can be correctly captured by its classifier vector $\mathbf{w}_\ell$. Moreover, the label classifier $\mathbf{w}_\ell$ is a function of the quality of the index as during training, the label classifiers are learned with negative sampling based on the search index. Therefore, how to improve the quality of the search index becomes a key challenge in the XMC problem.

There are two main formulations of the search index: 1) partition-based approach [25, 31, 7, 18, 32] and 2) approximate nearest neighbor search (ANNS) based approach [16, 9, 13, 10]. In partition-based approach, labels are first arranged into a tree-based index by partitioning the label space into mutually exclusive clusters and then a ML model is learned to route a given instance to a few relevant clusters. In an ANNS-based approach, a fixed, black-box ANNS index is learned on pre-defined label embeddings. Given an input embedding, this index is then used to efficiently query a small set of nearest labels based on some distance/similarity between the input and label embeddings. Both of these approaches suffer from a critical limitation that the index structure is fixed after it's initialized.

This decoupling of the search index from the rest of the ML model training prevents the search index from adapting with the rest of the model during training, which leads to sub-optimal performance.

To overcome this challenge, we propose a novel method called ELIAS: **E**nd-to-end **L**earning to **I**ndex **a**nd **S**earch, which jointly learns the search index along with the rest of the ML model for multi-label classification in large output spaces. In particular, as illustrated in Fig. 1, ELIAS generalizes the widely used partition tree-based index to a sparsely connected weighted graph-based index. ELIAS models the discrete cluster-to-label assignments in the existing partition based approaches as soft learnable parameters that are learned end-to-end with the encoder and classification module to optimize the final task objective. Moreover, because ELIAS uses a graph-based arrangement of labels instead of a tree-based arrangement, a label can potentially be assigned to multiple relevant clusters. This helps to better serve labels with a multi-modal input distribution [22].

Through extensive experiments we demonstrate that ELIAS achieves state-of-the-art results on multiple large-scale XMC benchmarks. Notably, ELIAS can be up to 2.5% better at precision@1 and up to 4% better at recall@100 than existing XMC methods. ELIAS's search index can be efficiently implemented on modern GPUs to offer fast inference times on million scale datasets. In particular, ELIAS offers sub-millisecond prediction latency on a dataset with 3 million labels on a single GPU.

## 2   Related Work

**One-vs-all (OVA) methods**: OVA methods consider classification for each label as an independent binary classification problem. In particular, an OVA method learns $L$ (number of classes) independent label classifiers $[\mathbf{w}_\ell]_{\ell=1}^L$, where the job of each classifier $\mathbf{w}_\ell$ is to distinguish training points of label $\ell$ from the rest of the training points. At prediction time, each label classifier is evaluated and the labels are ranked according to classifier scores. Traditional OVA methods like DiSMEC [2], ProXML [3], and PPDSparse [30] represent each input instance by their sparse bag-of-word features and learn

sparse linear classifiers by massively parallelizing over multiple machines. OVA methods achieve promising results on XMC benchmarks but suffer from huge computational complexity because of their linear scaling with number of labels. Subsequent XMC methods borrow the same building blocks of an OVA approach but overcome the computational overhead by employing some form of search index to efficiently shortlist only a few labels during training and prediction.

**Partition based methods**: Many XMC methods such as Parabel [25], Bonsai [19], XR-Linear [32], AttentionXML [31], X-Transformer [28], XR-Transformer [33], LightXML [18] follow this approach where the label space is partitioned into a small number of mutually exclusive clusters, and then an ML model is learned to route a given instance to a few relevant clusters. A popular way to construct clusters is to perform balanced $k$-means clustering recursively using some pre-defined input features. Traditional methods like Parabel, Bonsai, and XR-Linear represent their input by sparse bag-of-word features and learn sparse linear classifiers with negative sampling performed based on the search index. With the advancement of deep learning in NLP, recent deep learning based XMC methods replace sparse bag-of-word input features with dense embeddings obtained from a deep encoder. In particular, AttentionXML uses a BiLSTM encoder while X-Transformer, XR-Transformer, and LightXML use deep transformer models such as BERT [11] to encode the raw input text. In addition to dense embeddings, the state-of-the-art XR-Transformer method uses a concatenation of dense embedding and sparse bag-of-word features to get a more elaborate representation of the input, thus mitigating the information loss in text truncation in transformers.

**ANNS based methods**: Methods like SLICE [16], DeepXML [9], and GLaS [13] utilize approximate nearest neighbor search (ANNS) structure over pre-defined label representations to efficiently shortlist labels. In particular, SLICE represents each input instance by its FastText [24] embedding and uses the mean of a label's training points as a surrogate embedding for that label. It further constructs an HNSW [23] graph (a popular ANNS method) over these surrogate label embeddings. For a given input, the HNSW graph is queried to efficiently retrieve nearest indexed labels based on the cosine similarity between the input and label embedding. DeepXML extends SLICE by learning an MLP text encoder on a surrogate classification task instead of using a fixed FastText model to obtain input embeddings. GLaS takes a different approach and learns label classifiers with random negative sampling. After the model is trained, it constructs an ANNS index to perform fast maximum inner product search (MIPS) directly on the learned label classifiers.

**Learning search index**: There has been prior works [21, 1] that model different types of standard data structures with neural networks. A recent paper [27] models the search index in information retrieval systems as a sequence to sequence model where all the parameters of the search index is encoded in the parameters of a big transformer model. In a more similar spirit to our work, another recent paper [22] attempts to learn overlapping cluster partitions for XMC tasks by assigning each label to multiple clusters. Even though it serves as a generic plug-in method to improve over any existing partition based XMC method, it still suffers from the following shortcomings: 1) label assignments are not learned end-to-end with the task objective; instead, it alternates between finding the right model given the fixed label assignments and then finding the right label assignments given the fixed model, 2) all labels are assigned to a pre-defined number of clusters with equal probability and get duplicated in each assigned cluster, which results in increased computational complexity of the method.

## 3   ELIAS: End-to-end Learning to Index and Search

The multi-label classification problem can be formulated as following: given an input $\mathbf{x} \in \mathcal{X}$, predict $\mathbf{y} \in \{0, 1\}^L$ where $\mathbf{y}$ is a sparse $L$ dimensional vector with $y_\ell = 1$ if and only if label $\ell$ is relevant to input $\mathbf{x}$. Here, $L$ denotes the number of distinct labels - note that $\mathbf{y}$ can have multiple non-zero entries resulting in multiple label assignments to input $\mathbf{x}$. The training dataset is given in the form of $\{(\mathbf{x}^i, \mathbf{y}^i) : i = 1, ..., N\}$. XMC methods address the case where the label space ($L$) is extremely large (in the order of few hundred thousands to millions). All deep learning based XMC methods have the following three key components:

**Deep encoder** $\phi : \mathcal{X} \rightarrow \mathbb{R}^D$ which maps the input $\mathbf{x}$ to a $D$-dimensional dense embedding through a differentiable function. For text input, a popular choice of $\phi$ is the BERT [11] encoder where each input $\mathbf{x}$ is represented as a sequence of tokens.

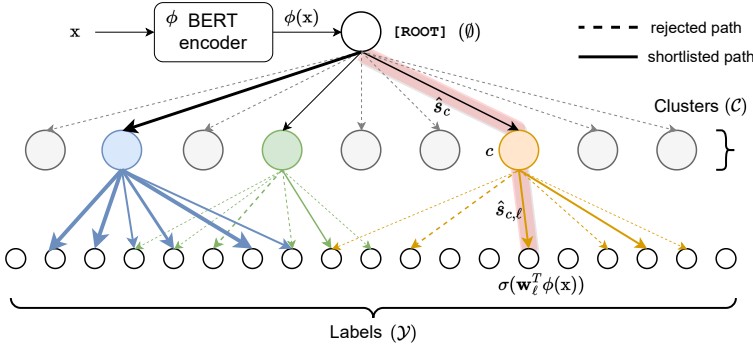

Figure 2: Illustration of ELIAS's search procedure: an input $\mathbf{x}$ is first embedded by the text encoder $\phi$ to get its embedding $\phi(\mathbf{x})$. Only a few (beam-size) clusters are shortlisted based on cluster relevance scores $\hat{s}_c \sim \hat{\mathbf{w}}_c^T \phi(\mathbf{x})$. All potential edges of shortlisted clusters are explored and assigned a score based on the product $\hat{s}_c * \hat{s}_{c,\ell}$ ($\hat{s}_{c,\ell}$ is normalized form of learnable edge weight parameter $a_{c,\ell}$ between cluster $c$ and label $l$). Top-$K$ paths are shortlisted based on their assigned scores and the final label relevance is computed as $\sigma(\mathbf{w}_\ell^T \phi(x)) * \hat{s}_{c,\ell} * \hat{s}_c$, here $\sigma$ is the sigmoid function. If a label $\ell$ can be reached from multiple paths then the path with maximal score is kept and rest are discarded.

**Search Index** $\mathcal{I} : \mathcal{X} \to \mathbb{R}^L$ shortlists $K$ labels along with a score assigned to each shortlisted label for a given input $\mathbf{x}$. More specifically, $\hat{\mathbf{y}} = \mathcal{I}(\mathbf{x})$ is a sparse real valued vector with only $K$ ($\ll L$) non-zero entries and $\hat{y}_\ell \neq 0$ implies that label $\ell$ is shortlisted for input $\mathbf{x}$ with shortlist relevance score $\hat{y}_\ell$. As illustrated in Figure 1, many partition based methods [18, 33] formulate their index as a label tree derived by hierarchically partitioning the label space into $C$ clusters and then learn classifier vectors $\hat{\mathbf{W}}_C = [\hat{\mathbf{w}}_c]_{c=1}^C$ ($\hat{\mathbf{w}}_c \in \mathbb{R}^D$) for each cluster which is used to select only a few clusters for a given input. More specifically, given the input $\mathbf{x}$, the relevance of cluster $c$ to input $\mathbf{x}$ is quantified by *cluster relevance scores* $\hat{s}_c = \hat{\mathbf{w}}_c^T \phi(\mathbf{x})$. The top-$b$ clusters based on these scores are selected and all labels inside the shortlisted clusters are returned as the shortlisted labels, where $b(\ll C)$ is a hyperparameter denoting the beam-size.

**Label classifiers** $\mathbf{W}_L = [\mathbf{w}_\ell]_{\ell=1}^L$ where $\mathbf{w}_\ell \in \mathbb{R}^D$ represents the classifier vector for label $\ell$ and $\mathbf{w}_\ell^T \phi(\mathbf{x})$ represents the *label relevance score* of label $\ell$ for input $\mathbf{x}$. As explained above, $\mathbf{w}_\ell^T \phi(\mathbf{x})$ is only computed for a few shortlisted labels obtained from the search index $\mathcal{I}$.

### 3.1 ELIAS Index

ELIAS formulates its label index as a specialized weighted graph between a root node $\emptyset$, $C$ cluster nodes $\mathcal{C} = \{c\}_{c=1}^C$ and $L$ label nodes $\mathcal{Y} = \{\ell\}_{\ell=1}^L$. As illustrated in Figure 2, all cluster nodes are connected to the root node and all label nodes are sparsely connected to few cluster nodes. ELIAS parameterizes the cluster-to-label edge assignments by a learnable adjacency matrix $\mathbf{A} = [a_{c,\ell}]_{C \times L}$, where the scalar parameter $a_{c,\ell}$ denotes the edge importance between cluster $c$ and label $\ell$.

Note that $\mathbf{A}$ can be very large for XMC datasets and using a dense $\mathbf{A}$ will incur $\mathcal{O}(CL)$ cost in each forward pass which can be computationally prohibitive for large-scale datasets. To mitigate this we restrict $\mathbf{A}$ to be a row-wise sparse matrix i.e. $\|\mathbf{a}_i\|_0 \leq \kappa$ where $\|.\|_0$ represents the $\ell_0$ norm, $\mathbf{a}_i$ represents the $i^{th}$ row of $\mathbf{A}$ and $\kappa$ is a hyper-parameter which controls the sparsity of $\mathbf{A}$. During training, only the non-zero entries of $\mathbf{A}$ is learned and the zero entries do not participate in any calculation. We defer the details of how we initialize the sparsity structure of $\mathbf{A}$ to Section 3.4.

Existing partition based XMC methods can be thought of as a special case of this formulation by adding additional restrictions that 1) each label is connected to exactly one cluster node, and 2) all cluster-to-label connections have equal importance. Moreover, existing methods initialize the cluster-to-label adjacency matrix $\mathbf{A}$ beforehand based on clustering over pre-defined features and keep it fixed throughout the training procedure. ELIAS overcomes these shortcomings by enabling the model to learn the cluster-to-label edge importance.

### 3.2 Forward Pass

ELIAS trains the entire model, including the deep encoder $\phi$, the search index parameters $\hat{\mathbf{W}}_C$, $\mathbf{A}$ and the label classifiers $\mathbf{W}_L$ in an end-to-end manner. We now describe the details of the forward pass of ELIAS.

**Text representation**: An input $\mathbf{x}$ is embedded by the encoder $\phi$ into a dense vector representation $\phi(\mathbf{x})$. In particular, we use BERT-base [11] as the encoder and represent $\phi(\mathbf{x})$ by the final layer's CLS token vector.

**Query search index**: Recall that the goal of the search index $\mathcal{I}$ is to efficiently compute a shortlist of labels $\hat{\mathbf{y}} \in \mathbb{R}^L$, where $\hat{\mathbf{y}}$ is a sparse real valued vector with $K$ ($\ll L$) non-zero entries and $\hat{y}_\ell \neq 0$ implies that label $\ell$ is shortlisted for input $\mathbf{x}$ with shortlist score $\hat{y}_\ell$. Similar to existing methods, ELIAS achieves this by first shortlisting a small subset of clusters $\hat{\mathcal{C}} \subset \mathcal{C}$ based on cluster relevance scores defined by $\hat{\mathbf{w}}_c^T \phi(\mathbf{x})$. But unlike existing methods which simply return the union of the fixed label set assigned to each shortlisted cluster, ELIAS shortlists the top-$K$ labels based on the soft cluster-to-label assignments and backpropagates the loss feedback to each of the shortlisted paths. More specifically, ELIAS defines the cluster relevance scores $\hat{\mathbf{s}}_\mathcal{C} \in \mathbb{R}^C$ as:

$$\hat{\mathbf{s}}_\mathcal{C} = [\hat{s}_c]_{c=1}^C = \min(1, \alpha * \text{softmax}(\hat{\mathbf{W}}_C^T \phi(\mathbf{x}))). \tag{1}$$

Here hyperparameter $\alpha$ is multiplied by the softmax scores to allow multiple clusters to get high relevance scores. Intuitively, $\alpha$ controls how many effective clusters can simultaneously activate for a given input (in practice, we keep $\alpha \approx 10$).

Given cluster relevance scores $\hat{\mathbf{s}}_\mathcal{C}$, we define set $\mathcal{C}_{topb}$ as the top $b$ clusters with the highest cluster relevance scores, where $b$ ($\ll C$) is the beam size hyperparameter. In the training phase, we further define a parent set $\mathcal{C}_{parent}$ to guarantee that the correct labels of $\mathbf{x}$ are present in the shortlist. More specifically, for each positive label of $\mathbf{x}$, we include the cluster with the strongest edge connection to $l$ in $\mathcal{C}_{parent}$. The shortlisted set $\hat{\mathcal{C}}$ is defined as the union of these two sets and the selection process can be summarized as follows:

$$\mathcal{C}_{topb} = \arg \text{top-}b(\hat{\mathbf{s}}_\mathcal{C}), \text{where } b(\ll C) \text{ is the beam size}, \tag{2}$$

$$\mathcal{C}_{parent} = \begin{cases} \{\} & \text{during prediction,} \\ \bigcup_{\ell:y_\ell=1}\{\arg \max_c(a_{c,\ell})\} & \text{during training} \end{cases} \tag{3}$$

$$\hat{\mathcal{C}} = \mathcal{C}_{topb} \cup \mathcal{C}_{parent}. \tag{4}$$

After shortlisting a small subset of clusters $\hat{\mathcal{C}}$, all potential edges of shortlisted clusters are explored and a set $\hat{\mathcal{P}}$ of explored paths is constructed, where $\hat{\mathcal{P}} = \{\emptyset \rightarrow c \rightarrow \ell : c \in \hat{\mathcal{C}} \text{ and } a_{c,\ell} > 0\}$. Furthermore, each path $\emptyset \rightarrow c \rightarrow \ell \in \hat{\mathcal{P}}$ is assigned a path score $\hat{s}_{\emptyset,c,\ell}$, where the path score $\hat{s}_{\emptyset,c,\ell}$ is expressed as the product of cluster relevance score $\hat{s}_c$ (defined by Eqn. 1) and edge score $\hat{s}_{c,\ell}$ which quantifies the probability of label $\ell$ getting assigned to cluster $c$ and is defined in terms of the learnable edge weight parameter $a_{c,\ell}$ as follows:

$$\hat{s}_{c,\ell} = \min(1, \beta * a_{c,\ell}^{norm}), \text{where } a_{c,\ell}^{norm} = \frac{\exp(a_{c,\ell})}{\sum_{\ell'=1}^L \exp(a_{c,\ell'})}, \text{and } \hat{s}_{\emptyset,c,\ell} = \hat{s}_c * \hat{s}_{c,\ell}. \tag{5}$$

Defining edge scores $\hat{s}_{c,\ell}$ in such a manner allows modelling the desired probability distribution of label assignment to a cluster, where a few relevant labels are assigned to a particular cluster with probability 1, and all other labels have probability 0. Hyperparameter $\beta$ controls how many effective labels can get assigned to a cluster, we choose $\beta \approx L/C$. Figure 3 empirically confirms that the trained model indeed learns the desired edge score distribution with most of the probability concentrated on a few labels and the rest of the labels getting assigned low probability. Moreover, this formulation also prevents labels with high softmax scores from overpowering edge assignments because as per 5, a relevant label $\ell$ for cluster $c$ gets positive feedback for $a_{c,\ell}$ only if $a_{c,\ell}^{norm} < 1/\beta$,

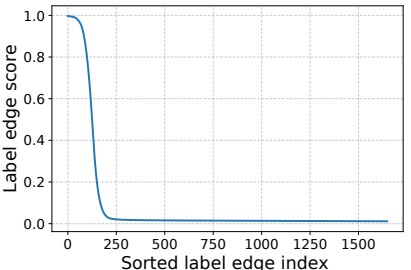

Figure 3: $a_{c,\ell}^{norm}$ (edge weight) distribution averaged over all clusters of trained ELIAS model on Amazon-670K dataset

otherwise $a_{c,\ell}$ does not participate in the calculation of $\hat{s}_{c,\ell}$. This allows clusters to learn balanced label assignments. Note that, because of the assumption that $\mathbf{A}$ is a row-wise sparse matrix, Eqn. 5 can be computed efficiently in $\mathcal{O}(\kappa)$ instead of $\mathcal{O}(L)$ time.

Since there can be multiple paths in $\hat{\mathcal{P}}$ which reach a particular label $\ell$, ELIAS defines shortlist score $\hat{y}_\ell$ for label $\ell$ by the maximum scoring path in $\hat{\mathcal{P}}$ that reaches $\ell$, i.e.

$$\hat{y}_\ell = \max_{c'} \{\hat{s}_{\emptyset,c',\ell} : \emptyset \to c' \to \ell \in \hat{\mathcal{P}}\}. \tag{6}$$

Finally, only the top-$K$ entries in $\hat{\mathbf{y}}$ are retained and the resulting vector is returned as the shortlist for input $\mathbf{x}$.

**Evaluating label classifiers**: label classifiers $[\mathbf{w}_\ell]_{\ell=1}^L$ are evaluated for the $K$ non-zero labels in $\hat{\mathbf{y}}$ and the final relevance score between label $\ell$ and input $\mathbf{x}$ is returned as $p_\ell = \sigma(\mathbf{w}_\ell^T \phi(\mathbf{x})) * \hat{y}_\ell$, here $\sigma$ is the sigmoid function.

## 3.3 Loss

ELIAS is trained on a combination of classification and shortlist loss where the shortlist loss encourages correct labels to have high shortlist scores ($\hat{y}_\ell$) and classification loss encourages positive labels in the shortlist to have high final score ($p_\ell$) and negative labels in the shortlist to have low final score. More specifically, the final loss $\mathcal{L}$ is defined as $\mathcal{L} = \mathcal{L}_c + \lambda \mathcal{L}_s$, where $\lambda$ is a hyperparameter and classification loss $\mathcal{L}_c$ is defined as binary cross entropy loss over shortlisted labels

$$\mathcal{L}_c = -\sum_{\ell:\hat{y}_\ell \neq 0} (y_\ell \log(p_\ell) + (1 - y_\ell)(1 - \log(p_\ell))), \tag{7}$$

shortlist loss $\mathcal{L}_s$ is defined as negative log likelihood loss over the positive labels

$$\mathcal{L}_s = -\sum_{\ell:y_\ell=1} \log(\hat{y}_\ell). \tag{8}$$

## 3.4 Staged Training

Previous sub-sections described the ELIAS framework for learning the index graph along with the ML model in an end-to-end manner. Although, in principle one can optimize the network with the given loss function from a random initialization but we highlight a few key challenges in doing so: 1) *Optimization challenge*: because of the flexibility in the network to assign a label node to various clusters, it becomes hard for a label to get confidently assigned to only a few relevant clusters. As a result, the model is always chasing a moving target and for a given input it is not able to be sure about any single path; 2) *Computational challenge*: the full cluster-label adjacency matrix $\mathbf{A}$ can be very large for large datasets and will incur $\mathcal{O}(CL)$ cost in each forward pass if implemented in dense form. To address these challenges we train the ELIAS model in two stages. In the first stage, we only train the encoder $\phi$, cluster classifiers $\hat{\mathbf{W}}_C$, and label classifiers $\mathbf{W}_L$ keeping $A$ fixed and assigned based on traditional balanced partitions. We then utilize the stage-1 trained model to initialize the sparse adjacency matrix $\mathbf{A}$. In the second stage, we take the initialized $\mathbf{A}$ and rest of the stage 1 model, and jointly train the full model $\phi, \hat{\mathbf{W}}_C, \mathbf{W}_L, \mathbf{A}$.

**Stage 1**: In stage 1 training, similar to existing partition-based XMC methods, we partition the label space into $C$ mutually exclusive clusters by performing balanced $k$-means clustering over pre-defined label features. The adjacency matrix induced by these clusters is then used as fixed assignment for $\mathbf{A}$. Keeping $\mathbf{A}$ fixed, we train the rest of the model (i.e. $\phi, \hat{\mathbf{W}}_C, \mathbf{W}_L$) on the loss described in Section 3.3. More details on clustering are provided in Section C.1 in the Appendix.

**Initializing $\mathbf{A}$**: As highlighted before, to overcome the $\mathcal{O}(CL)$ cost associated with a full adjacency matrix $\mathbf{A}$, we want to restrict $\mathbf{A}$ to be a row-wise sparse matrix. In other words, we want to restrict each cluster to choose from a candidate subset of $\kappa$ labels instead of the whole label set. Intuitively, in order for the model to learn anything meaningful, the candidate subset for each cluster should contain approximately similar labels. To achieve this, we utilize the stage 1 model to first generate an approximate adjacency matrix $\mathbf{A}'$ and then select the top-$\kappa$ entries in each row of $\mathbf{A}'$ as non-zero entries for $\mathbf{A}$. More specifically, we first identify top-$b$ matched clusters for each training point $\mathbf{x}^i$ by computing the cluster matching matrix $\mathbf{M} = [m_{i,c}]_{N \times C}$ as:

$$m_{i,c} = \begin{cases} \hat{\mathbf{s}}_c^i & \text{if } c \in \mathcal{C}_{topb}^i, \\ 0 & \text{otherwise} \end{cases} \tag{9}$$

where $\hat{\mathbf{s}}_c^i$ represents the cluster relevance score and $\mathcal{C}_{topb}^i$ represents the set of top-$b$ clusters for $i^{th}$ training point $\mathbf{x}^i$. After computing $\mathbf{M}$, we define the approximate adjacency matrix $\mathbf{A}' = [a'_{c,\ell}]_{C \times L} = \mathbf{M}^T \mathbf{Y}$, where $\mathbf{Y} = [\mathbf{y}^1, ..., \mathbf{y}^i, ..., \mathbf{y}^N]^T$. The element $a'_{c,\ell}$ essentially denotes the weighted count of how many times the cluster $c$ got placed in top-$b$ positions for positive training points of label $\ell$. Finally, the top $\kappa$ elements in each row of $\mathbf{A}'$ are selected as the non-zero parameters of $\mathbf{A}$, i.e.

$$a_{c,\ell} = \begin{cases} \texttt{random(0, 1)} & \text{if } \ell \in \arg \text{top-}\kappa(\mathbf{a}'_c) \\ 0 & \text{otherwise} \end{cases} \quad (10)$$

We choose a large enough $\kappa$ to provide the model enough degree of freedom to learn cluster-to-label assignments. In particular, $\kappa \sim 10 \times L/C$ works well across datasets without adding any computational burden. For efficient implementation on GPUs, we store matrix $\mathbf{A}$ in the form of two tensors, one storing the non-zero indices and the other storing the values corresponding to those non-zero indices.

**Stage 2**: In stage 2 training, we initialize $\mathbf{A}$ as described above, and $\phi$, $\hat{\mathbf{W}}_C$ from stage 1 model. We then train the full ELIAS model (i.e. $\phi, \hat{\mathbf{W}}_C, \mathbf{W}_L, \mathbf{A}$) end-to-end to optimize the loss defined in Section 3.3.

### 3.5 Sparse Ranker

State-of-the-art XMC methods like XR-Transformer [33] and X-Transformer [7] utilize high capacity sparse classifiers learned on the concatenated sparse bag-of-word features and dense embedding obtained from the deep encoder for ranking their top predictions. Because of the high capacity, sparse classifiers are able to represent head labels more elaborately than dense classifiers. Moreover, bag-of-words representation is able to capture the full input document instead of the truncated document that the deep encoder receives.

To compare fairly with such methods, we explore an enhanced variant of ELIAS represented by ELIAS ++, which additionally learns a sparse ranker that re-ranks the top 100 predictions of ELIAS. In particular, the sparse ranker takes the concatenated sparse bag-of-word and dense embedding input features and learns sparse linear classifiers on the top 100 label predictions made from the trained ELIAS model for each training point. Because these sparse classifiers are only trained on 100 labels per training point, they can be quickly trained by parallel linear solvers like LIBLINEAR [12]. We use the open-source PECOS[2] [32] library to train and make predictions with the sparse ranker.

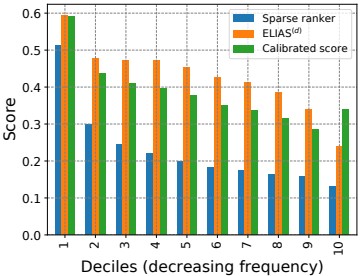

Figure 4: True label's score distribution of sparse ranker and ELIAS $^{(d)}$ over different label deciles on Amazon-670K dataset. $1^{st}$ decile represents labels with most training points while $10^{th}$ decile represents labels with least training points

During prediction, the top 100 predictions are first made by ELIAS and then the learned sparse ranker is evaluated on these top 100 predictions. We empirically observe that the scores returned by ELIAS and sparse ranker are not well calibrated across different label regimes. As shown in Figure 4, the sparse ranker underestimates scores on tail labels while ELIAS scores are more balanced across all label regimes. To correct this score mis-calibration, we learn a simple score calibration module which consists of a standard decision tree classifier[3] that takes both of these scores and the training frequency of the label as input and predicts a single score denoting the label relevance. The score calibration module is learned on a small validation set of 5,000 points. More details on the sparse ranker are in Appendix Section C.2.

### 3.6 Time Complexity Analysis

The time complexity for processing a batch of $n$ data-points is $\mathcal{O}(n(T_{\text{bert}} + Cd + b\kappa + Kd))$ where $T_{\text{bert}}$ represents the time complexity of the bert encoder, $C$ represents the number of clusters in index,

---

[2]https://github.com/amzn/pecos
[3]https://scikit-learn.org/stable/modules/generated/sklearn.tree.DecisionTreeClassifier.html

Table 1: Performance comparison on extreme classification benchmark datasets. Bold numbers represent overall best numbers for that dataset while underlined numbers represent best numbers for dense embedding based methods. Methods which only use sparse bag-of-word features are distinguished by $^{(s)}$ superscript, dense embedding based methods are distinguished by $^{(d)}$ superscript and methods that use both sparse + dense features are distinguished by $^{(s+d)}$ superscript

| Method | P@1 | P@3 | P@5 | PSP@1 | PSP@3 | PSP@5 | P@1 | P@3 | P@5 | PSP@1 | PSP@3 | PSP@5 |
|---|---|---|---|---|---|---|---|---|---|---|---|---|
| | | | Amazon-670K | | | | | | LF-AmazonTitles-131K | | | |
| DiSMEC$^{(s)}$ | 44.70 | 39.70 | 36.10 | 27.80 | 30.60 | 34.20 | 35.14 | 23.88 | 17.24 | 25.86 | 32.11 | 36.97 |
| Parabel$^{(s)}$ | 44.89 | 39.80 | 36.00 | 25.43 | 29.43 | 32.85 | 32.60 | 21.80 | 15.61 | 23.27 | 28.21 | 32.14 |
| XR-Linear$^{(s)}$ | 45.36 | 40.35 | 36.71 | - | - | - | 34.11 | 23.06 | 16.63 | 24.75 | 30.35 | 34.86 |
| Bonsai$^{(s)}$ | 45.58 | 40.39 | 36.60 | 27.08 | 30.79 | 34.11 | 34.11 | 23.06 | 16.63 | 24.75 | 30.35 | 34.86 |
| Slice$^{(d)}$ | 33.15 | 29.76 | 26.93 | 20.20 | 22.69 | 24.70 | 30.43 | 20.50 | 14.84 | 23.08 | 27.74 | 31.89 |
| Astec$^{(d)}$ | 47.77 | 42.79 | 39.10 | 32.13 | 35.14 | 37.82 | 37.12 | 25.20 | 18.24 | 29.22 | 34.64 | 39.49 |
| GLaS$^{(d)}$ | 46.38 | 42.09 | 38.56 | **38.94** | **39.72** | 41.24 | - | - | - | - | - | - |
| AttentionXML$^{(d)}$ | 47.58 | 42.61 | 38.92 | 30.29 | 33.85 | 37.13 | 32.55 | 21.70 | 15.64 | 23.97 | 28.60 | 32.57 |
| LightXML$^{(d)}$ | 49.10 | 43.83 | 39.85 | - | - | - | 38.49 | 26.02 | 18.77 | 28.09 | 34.65 | 39.82 |
| XR-Transformer$^{(s+d)}$ | 50.11 | 44.56 | 40.64 | 36.16 | 38.39 | 40.99 | 38.42 | 25.66 | 18.34 | 29.14 | 34.98 | 39.66 |
| Overlap-XMC$^{(s+d)}$ | 50.70 | 45.40 | 41.55 | 36.39 | 39.15 | **41.96** | - | - | - | - | - | - |
| ELIAS $^{(d)}$ | _50.63_ | _45.49_ | _41.60_ | 32.59 | 36.44 | 39.97 | _39.14_ | _26.40_ | _19.08_ | _30.01_ | _36.09_ | _41.07_ |
| ELIAS ++$^{(s+d)}$ | **53.02** | **47.18** | **42.97** | 34.32 | 38.12 | 41.93 | **40.13** | **27.11** | **19.54** | **31.05** | **37.57** | **42.88** |
| | | | Wikipedia-500K | | | | | | Amazon-3M | | | |
| DiSMEC$^{(s)}$ | 70.21 | 50.57 | 39.68 | 31.20 | 33.40 | 37.00 | 47.34 | 44.96 | 42.80 | - | - | - |
| Parabel$^{(s)}$ | 68.70 | 49.57 | 38.64 | 26.88 | 31.96 | 35.26 | 47.48 | 44.65 | 42.53 | 12.82 | 15.61 | 17.73 |
| XR-Linear$^{(s)}$ | 68.12 | 49.07 | 38.39 | - | - | - | 47.96 | 45.09 | 42.96 | - | - | - |
| Bonsai$^{(s)}$ | 69.20 | 49.80 | 38.80 | - | - | - | 48.45 | 45.65 | 43.49 | 13.79 | 16.71 | 18.87 |
| Slice$^{(d)}$ | 62.62 | 41.79 | 31.57 | 24.48 | 27.01 | 29.07 | - | - | - | - | - | - |
| Astec$^{(d)}$ | 73.02 | 52.02 | 40.53 | 30.69 | 36.48 | 40.38 | - | - | - | - | - | - |
| GLaS$^{(d)}$ | 69.91 | 49.08 | 38.35 | - | - | - | - | - | - | - | - | - |
| AttentionXML$^{(d)}$ | 76.95 | 58.42 | 46.14 | 30.85 | 39.23 | 44.34 | 50.86 | 48.04 | 45.83 | 15.52 | 18.45 | 20.60 |
| LightXML$^{(d)}$ | 77.78 | 58.85 | 45.57 | - | - | - | - | - | - | - | - | - |
| XR-Transformer$^{(s+d)}$ | 79.40 | 59.02 | 46.25 | **35.76** | 42.22 | 46.36 | 54.20 | 50.81 | 48.26 | **20.52** | **23.64** | **25.79** |
| Overlap-XMC$^{(s+d)}$ | - | - | - | - | - | - | 52.70 | 49.92 | 47.71 | 18.79 | 21.90 | 24.10 |
| ELIAS $^{(d)}$ | _79.00_ | _60.37_ | _46.87_ | _33.86_ | _42.99_ | _47.29_ | _51.72_ | _48.99_ | _46.89_ | _16.05_ | _19.39_ | _21.81_ |
| ELIAS ++$^{(s+d)}$ | **81.26** | **62.51** | **48.82** | 35.02 | **45.94** | **51.13** | **54.28** | **51.40** | **49.09** | 15.85 | 19.07 | 21.52 |

$d$ is the embedding dimension, $b$ is the beam size, $\kappa$ is the row-wise sparsity of cluster-to-label adjacency matrix $A$, and $K$ is the number of labels shortlisted for classifier evaluation. Assuming $C = \mathcal{O}(\sqrt{L})$, $\kappa = \mathcal{O}(L/C) = \mathcal{O}(\sqrt{L})$ and $K = \mathcal{O}(\sqrt{L})$, the final time complexity comes out to be $\mathcal{O}(n(T_{\text{bert}} + \sqrt{L}(2d + b)))$. Empirical prediction and training times on benchmark datasets are reported in Table 6 of the Appendix.

## 4   Experimental Results

**Experimental Setup** We conduct experiments on three standard full-text extreme classification datasets: Wikipedia-500K, Amazon-670K, Amazon-3M and one short-text dataset: LF-AmazonTitles-131K which only contains titles of Amazon products as input text. For Wikipedia-500K, Amazon-670K, and Amazon-3M, we use the same experimental setup (i.e. raw input text, sparse features and train-test split) as existing deep XMC methods [31, 33, 18, 7]. For LF-AmazonTitles-131K, we use the experimental setup provided in the extreme classification repository [5]. Comparison to existing XMC methods is done by standard evaluation metrics of precision@$K$ (P@$K = 1, 3, 5$) and its propensity weighted variant (PSP@$K = 1, 3, 5$) [15]. We also compare competing methods and baselines with ELIAS at recall@$K$ (R@$K = 10, 20, 100$) evaluation to illustrate the superior shortlisting performance of ELIAS's search index. More details on the experimental setup and dataset statistics are presented in Appendix Section B.

**Implementation details** Similar to existing XMC methods, we take an ensemble of 3 models with different initial clustering of label space to report final numbers. For efficient implementation on GPU, the raw input sequence is concatenated to 128 tokens for full-text datasets and 32 tokens for short-text dataset. Number of clusters $C$ for each dataset is chosen to be the same as LightXML which selects $C \sim L/100$. We keep the shortlist size hyperparameter $K$ fixed to 2000 which is approximately same as the number of labels existing partition based methods shortlist assuming beam-size $b = 20$

Table 2: Precision and recall comparison of single model dense embedding-based methods. ELIAS matches or even outperforms the brute-force Bert-OvA baseline while existing partition based methods fail to compare well, especially at recall values.

| Method | P@1 | P@3 | P@5 | R@10 | R@20 | R@100 | P@1 | P@3 | P@5 | R@10 | R@20 | R@100 |
|---|---|---|---|---|---|---|---|---|---|---|---|---|
| | | | Amazon-670K | | | | | | LF-AmazonTitles-131K | | | |
| BERT-OvA-1$^{(d)}$ | 48.50 | 43.41 | 39.67 | 49.53 | 56.60 | 67.90 | **38.17** | **25.66** | 18.44 | **50.29** | **54.71** | 62.80 |
| AttentionXML-1$^{(d)}$ | 45.84 | 40.92 | 37.24 | 45.59 | 51.25 | 60.77 | 30.26 | 20.03 | 14.31 | 38.16 | 41.47 | 47.73 |
| LightXML-1$^{(d)}$ | 47.29 | 42.24 | 38.48 | 47.34 | 53.26 | 62.03 | 37.01 | 24.88 | 17.90 | 48.07 | 52.10 | 59.42 |
| XR-Transformer-1$^{(d)}$ | 45.25 | 40.3 | 36.45 | 45.19 | 51.61 | 61.11 | 34.58 | 23.31 | 16.79 | 45.72 | 49.65 | 56.00 |
| ELIAS-1$^{(d)}$ | **48.68** | **43.78** | **40.04** | **50.33** | **57.67** | **68.95** | 37.90 | 25.61 | **18.45** | 50.12 | 54.62 | **62.88** |

Table 3: Performance analysis of different components of ELIAS. Allowing the model to learn cluster-to-label assignments significantly improves both precision and recall performance (see row 2 vs row 1). Sparse ranker further improves performance on top predictions (see row 4 vs row 2).

| Method | P@1 | P@3 | P@5 | R@10 | R@20 | R@100 | P@1 | P@3 | P@5 | R@10 | R@20 | R@100 |
|---|---|---|---|---|---|---|---|---|---|---|---|---|
| | | | Amazon-670K | | | | | | LF-AmazonTitles-131K | | | |
| Stage 1 | 46.63 | 41.65 | 37.58 | 46.08 | 52.29 | 61.72 | 36.96 | 24.67 | 17.69 | 47.69 | 51.74 | 58.81 |
| + Stage 2 | 48.68 | 43.78 | 40.04 | 50.33 | 57.67 | 68.95 | 37.90 | 25.61 | 18.45 | 50.12 | 54.62 | 62.88 |
| + Sparse ranker w/o calibration | 50.72 | 45.25 | 41.27 | 51.51 | 58.43 | 68.95 | 39.25 | 26.47 | 19.02 | 51.4 | 55.39 | 62.88 |
| + Score calibration | 51.41 | 45.69 | 41.62 | 51.97 | 58.81 | 68.95 | 39.26 | 26.47 | 19.02 | 51.4 | 55.35 | 62.88 |
| + 3× ensemble | 53.02 | 47.18 | 42.97 | 53.99 | 61.33 | 72.07 | 40.13 | 27.11 | 19.54 | 53.31 | 57.78 | 65.15 |

and the number of labels per cluster $= 100$. AdamW [20] optimizer is used to train the whole model with weight decay applied only to non-gain and non-bias parameters. Optimization update for label classifiers $\mathbf{W}_L$ is performed with high accumulation steps (i.e. optimization update is performed at every $k$ training steps, where $k = 10$) since updating $\mathbf{W}_L$ every step is a computational bottleneck and only few parameters inside $\mathbf{W}_L$ gets updated in each optimization step anyway. More details and hyperparameters for each dataset are presented in Appendix Section B.

**Comparison on XMC benchmarks** Table 1 compares our method with leading XMC methods such as DiSMEC [2], Parabel [25], XR-Linear [32], Bonsai [19], Slice [16], Astec [9], GlaS [13], AttentionXML [31], LightXML [18], XR-Transformer [33], and Overlap-XMC [22]. Most baseline results are obtained from their respective papers when available and otherwise taken from results reported in [31, 33] and extreme classification repository [5]. To allow fair comparison among methods that use the same form of input representation, we distinguish methods that use only sparse bag-of-word input features by $^{(s)}$ superscript, methods that use only dense embedding based input features by $^{(d)}$ superscript, and methods that use both sparse + dense features by $^{(s+d)}$ superscript. ELIAS ++ which uses sparse + dense features achieves state-of-the-art performance on all datasets at precision values while being either the best or second best method at propensity scored precision on most datasets. The dense embedding based ELIAS $^{(d)}$ consistently outperforms existing dense embedding based XMC methods by significant margin and on many occasions achieves gains over previous state-of-the-art methods which use both sparse + dense features.

**Comparison with brute-force OvA baseline** To establish the classification performance that could have been achieved if there was no sampling performed by the shortlisting procedure, we implement the brute-force one-versus-all baseline BERT-OvA which consists of BERT encoder followed by a fully connected linear classification layer and is trained and inferred in one-versus-all fashion without any sampling. We follow the same training procedures as ELIAS for this baseline. Table 2 compares the OvA baseline with ELIAS and leading deep XMC methods such as AttentionXML, LightXML and a dense version of XR-Transformer which uses only dense embeddings, under single model (i.e. no ensemble) setup for direct comparison. Existing deep XMC methods do not compare well against the OvA model especially at recall@100 but ELIAS matches and sometimes even marginally outperforms, the brute-force OvA baseline while enjoying faster training and inference speed due to the search index.

**Component wise ablation of ELIAS** Table 3 presents a build-up ablation of performance gains made by different components of ELIAS. The stage 1 model which fixes its adjacency matrix by clustering labels into mutually exclusive clusters performs similarly to existing single model XMC methods. Allowing the model to learn the adjacency matrix $\mathbf{A}$ in stage 2 improves recall by up to 7% and precision by up to 2.5% over the stage 1 model. Adding the sparse ranker and score-calibration

module further improves model performance on top predictions but the gains diminish as we increase prediction set size. Finally, the ensemble of 3 models improves performance at all evaluation metrics which is a well observed behaviour with all XMC methods.

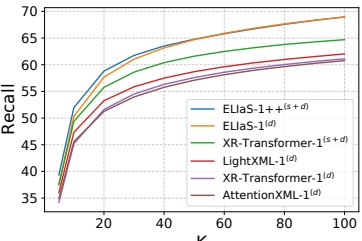 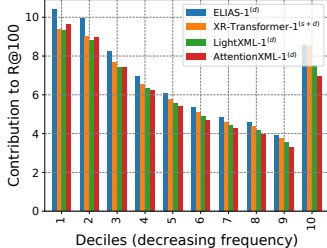

Figure 5: (*left*) Comparison of recall at different prediction set size on Amazon-670K (*right*) Decile-wise analysis of recall@100 on Amazon-670K, $1^{st}$ decile represents labels with most training points i.e. head labels while $10^{th}$ decile represents labels with least training points i.e. tail labels

**Recall comparison** Next, we compare the recall capabilities of existing methods with ELIAS. The left plot in Figure 5 plots the recall at different prediction set size for all competing methods and ELIAS. ELIAS strictly outperforms existing methods at all prediction set sizes and in particular, can be up to 4% better at recall@100 than the next best method. To further investigate which label regimes benefit most from ELIAS's search index we plot the decile wise contribution to recall@100 for each method. As we can see, ELIAS improves recall performance over existing methods in each label decile but the most improvement come from the top 2 deciles representing the most popular labels. We hypothesize that because the popular labels are likely to have multi-modal input distribution, existing partition based methods which assign a label to only one cluster fail to perform well on these multi-modal labels. Section C.4 contains additional discussion and results to support this claim.

## 5 Conclusion and Discussions

In this paper, we propose ELIAS, which extends the widely used partition tree based search index to a learnable graph based search index for extreme multi-label classification task. Instead of using a fixed search index, ELIAS relaxes the discrete cluster-to-label assignments in the existing partition based approaches as soft learnable parameters. This enables the model to learn a flexible index structure, and it allows the search index to be learned end-to-end with the encoder and classification module to optimize the final task objective. Empirically, ELIAS achieves state-of-the-art performance on several large-scale extreme classification benchmarks with millions of labels. ELIAS can be up to 2.5% better at precision@1 and up to 4% better at recall@100 than existing XMC methods.

This work primarily explores many-shot and few-shot scenarios where some training supervision is available for each label (output). It would be interesting to see how we can adapt the proposed solution to zero-shot scenarios where there is no training supervision available for the labels. One potential approach could be to parameterize the cluster-to-label adjacency matrix as a function of cluster and label features instead of free learnable scalars. Furthermore, one limitation of the proposed solution is that it learns a shallow graph structure over label space; this may not be ideal for scaling the method to billion-scale datasets. It would be exciting to explore how one can extend ELIAS to learn deep graph structures.

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
