# A  Potential Negative Societal Impact

Our method proposes to learn efficient data structure for accurate prediction in large-output space. It helps existing large-scale retrieval systems used in various online applications to efficiently produce more accurate results. To the best of our knowledge, this poses no negative impacts on society.

# B  Experimental Details

## B.1  ELIAS Hyperparameters

ELIAS's hyperparameters include,

- `max-len`: denotes the maximum sequence length of input for the BERT encoder. As per standard XMC practices, for full-text dataset we choose 128 and for short-text we choose 32

- $C$: denotes number of clusters in the index graph, we use same values as LightXML [18] and X-Transformer [6] for fair comparison

- $\alpha$: multiplicative hyperparameter used in Equation 1, controls effective number of clusters that can get activated for a given input

- $\beta$: multiplicative hyperparameter used in Equation 5, controls effective number of labels that can get assigned to a particular cluster

- $\kappa$: controls the row-wise sparsity of adjacency matrix $\mathbf{A}$, we choose $\kappa \approx 10 \times L/C$

- $\lambda$: controls importance of classification loss $\mathcal{L}_c$ and shortlist loss $\mathcal{L}_s$ in the final loss $\mathcal{L}$, we choose $\lambda$ by doing grid search over the smallest dataset LF-AmazonTitles-131K

- $K$: denotes the shortlist size, label classifiers are only evaluated on top-$K$ shortlisted labels. We choose $K = 2000$ which is approximately same as the number of labels existing partition based methods shortlist assuming beam-size $b = 20$ and number of labels per cluster $= 100$

- $b$: denotes the beam size, similar to existing partition based methods we use $b = 20$

- `num-epochs`: denotes the total number of epochs (i.e. including stage 1 and stage 2 training)

- $LR_{\mathbf{W}}, LR_{\phi}$: We empirically observe that the network trains faster when we decouple the initial learning rates of the transformer encoder ($LR_{\phi}$) with rest of the model ($LR_{\mathbf{W}}$). We choose a much smaller values for $LR_{\phi}$ and a relatively larger value for $LR_{\mathbf{W}}$

- `bsz`: denotes the batch-size of the mini-batches used during training

Table 4: ELIAS hyperparameters

| Dataset | max-len | $C$ | $\alpha$ | $\beta$ | $\kappa$ | $\lambda$ | $K$ | $b$ | num-epochs | $LR_{\mathbf{W}}$ | $LR_{\phi}$ | bsz |
|---|---|---|---|---|---|---|---|---|---|---|---|---|
| LF-AmazonTitles-131K | 32 | 2048 | 10 | 150 | 1000 | 0.05 | 2000 | 20 | 60 | 0.02 | $1e^{-4}$ | 512 |
| Amazon-670K | 128 | 8192 | 10 | 150 | 1000 | 0.05 | 2000 | 20 | 60 | 0.01 | $1e^{-4}$ | 256 |
| Wikipedia-500K | 128 | 8192 | 10 | 150 | 1000 | 0.05 | 2000 | 20 | 45 | 0.005 | $5e^{-5}$ | 256 |
| Amazon-3M | 128 | 32768 | 20 | 150 | 1000 | 0.05 | 2000 | 20 | 45 | 0.002 | $2e^{-5}$ | 64 |

## B.2  Datasets

**LF-AmazonTitles-131K**: A product recommendation dataset where input is the title of the product and labels are other related products to the given input. "LF-*" datasets additionally contain label features i.e. a label is not just an atomic id, label features which describe a label are also given. For this paper, we don't utilize these additional label features and compare ELIAS to only methods which don't utilize label features either. Notably, even though ELIAS doesn't use label features it achieves very competitive performance with methods which use the label features in their model.

**Amazon-670K**: A product recommendation dataset where input is a textual description of a query product and labels are other related products for the query.

**Wikipedia-500K**: A document tagging dataset where input consists of full text of a wikipedia page and labels are wikipedia tags relevant to that page.

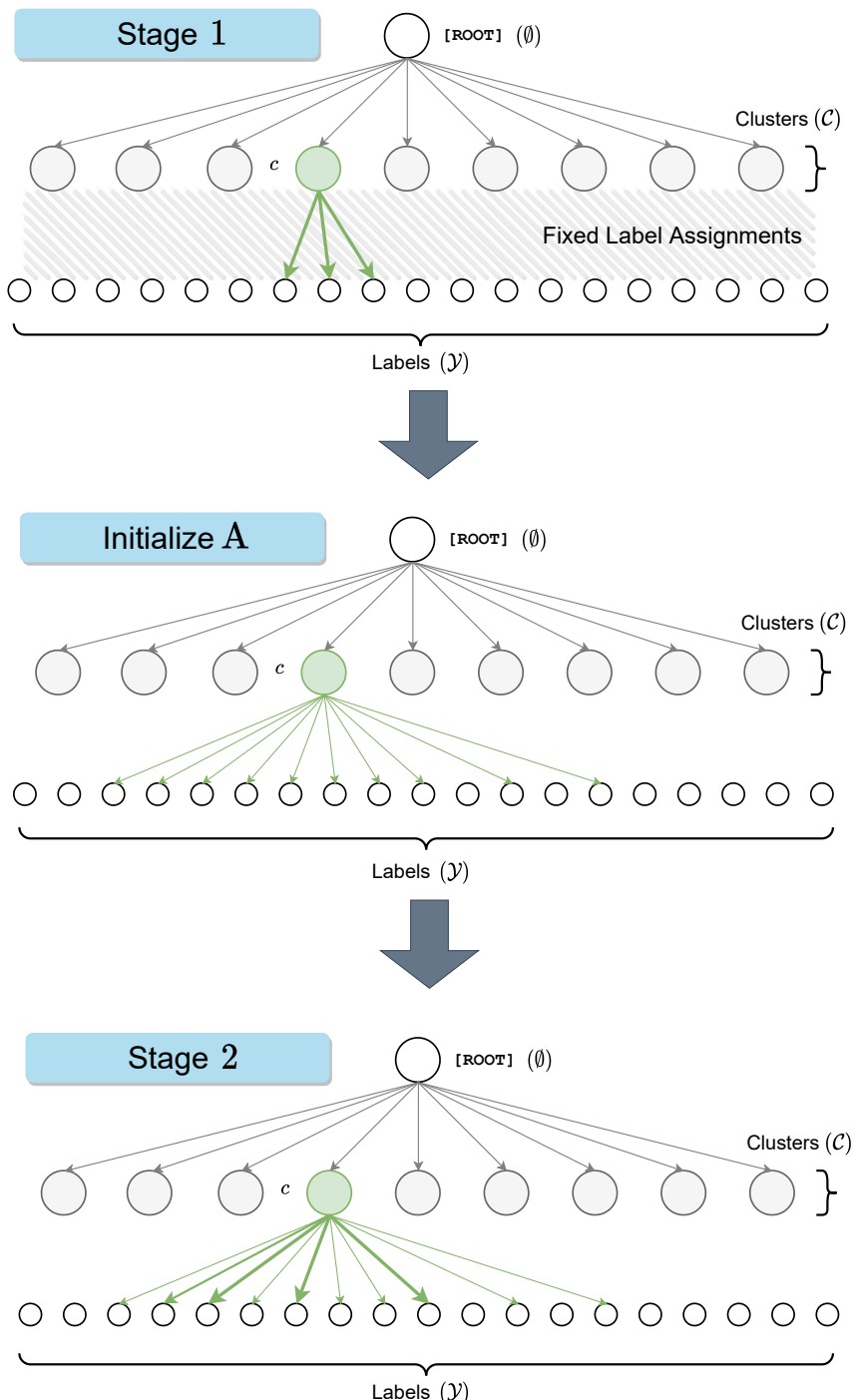

Figure 6: Illustration of ELIAS's search index graph in different training stages.

**Amazon-3M**: A product recommendation dataset where input is a textual description of a query product and labels are other co-purchased products for the query.

## B.3 Evaluation Metrics

We use standard Precision@$K$ (P@$K$), propensity weighted variant of Precision (PSP@$K$), and Recall@$K$ (R@$K$) evaluation metrics for comparing ELIAS to baseline methods. For a single

Table 5: Dataset statistics, here $D_{\text{bow}}$ denotes the dimensionality of sparse bag-of-word features

| Dataset | Num Train Points | Num Test Points | Num Labels | Avg. Labels per Point | Avg. Points per Label | $D_{\text{bow}}$ |
|---|---|---|---|---|---|---|
| LF-AmazonTitles-131K | 294,805 | 134,835 | 131,073 | 2.29 | 5.15 | 40,000 |
| Amazon-670K | 490,449 | 153,025 | 670,091 | 3.99 | 5.45 | 135,909 |
| Wikipedia-500K | 1,779,881 | 769,421 | 501,070 | 4.75 | 16.86 | 2,381,304 |
| Amazon-3M | 1,717,899 | 742,507 | 2,812,281 | 22.02 | 36.06 | 337,067 |

data-point $i$, these evaluation metrics can be formally defined as:

$$\text{P@}K = \frac{1}{K} \sum_{j=1}^{K} y_{rank(j)}^{i} \tag{11}$$

$$\text{PSP@}K = \sum_{j=1}^{K} \frac{y_{rank(j)}^{i}}{p_{rank(j)}} \tag{12}$$

$$\text{R@}K = \frac{1}{\|\mathbf{y}^i\|_0} \sum_{j=1}^{K} y_{rank(j)}^{i} \tag{13}$$

Where, $\mathbf{y}^i = [y_l^i]_{l=1}^{L}, y_l \in \{0, 1\}$ represents the ground truth label vector, $p$ represents the propensity score vector [17], $\|.\|_0$ represents the $\ell_0$ norm, and $rank(j)$ denotes the index of j$^{th}$ highest ranked label in prediction vector of input $i$.

## C More on ELIAS

### C.1 Additional Training Details

Figure 6 illustrates the evolution of ELIAS's search index graph over different stages of training. In stage 1, label to cluster assignments are pre-determined and fixed by clustering all labels into $C$ clusters. Then, rest of the ML model i.e. $\phi, \mathbf{W}_C, \mathbf{W}_L$ is trained. The model obtained after stage 1 training is used to initialize the row-wise sparse adjacency matrix $\mathbf{A}$ as described in Section 3.4. In stage 2, the non-zero entries in the sparse adjacency matrix $\mathbf{A}$ along with the rest of the ML model is trained jointly to optimize the task objective. The clustering procedure used in stage 1 can be described as follows:

We first obtain a static representation $\psi(\mathbf{x}^i)$ for each training point $\mathbf{x}^i$ as:

$$\psi(\mathbf{x}^i) = \left[\frac{\text{bow}(\mathbf{x}^i)}{\|\text{bow}(\mathbf{x}^i)\|_2}, \frac{\phi(\mathbf{x}^i)}{\|\phi(\mathbf{x}^i)\|_2}\right] \tag{14}$$

Here, $[\,]$ represents the concatenation operator, $\text{bow}(\mathbf{x}^i)$ represents the sparse bag-of-words representation of $\mathbf{x}^i$ and $\phi(\mathbf{x}^i)$ represents the deep encoder representation of $\mathbf{x}^i$. Next we define label centroids $\mu_l$ for each label as:

$$\mu_l = \frac{\sum_{i:y_l^i=1} \psi(\mathbf{x}^i)}{\|\sum_{i:y_l^i=1} \psi(\mathbf{x}^i)\|_2} \tag{15}$$

We then cluster all labels into $C$ clusters by recursively performing balanced 2-means [25] over label centroids $\{\mu_l\}_{l=1}^{L}$. This gives us a clustering matrix $\mathbf{C} \in \mathbb{R}^{C \times L}$, where $\mathbf{C}_{c,l} = 1$ iff label $l$ got assigned to cluster $c$. Note that, a label is assigned to only one cluster and each cluster gets assigned equal number of labels. We assign this clustering matrix $\mathbf{C}$ to the label-cluster adjacency matrix $\mathbf{A}$ and keep it frozen during the stage 1 training i.e. only parameters $\phi, \mathbf{W}_C, \mathbf{W}_L$ are trained on the loss defined in Section 3.3.

### C.2 Additional Sparse Ranker Details

In this subsection we describe the training and prediction procedure of sparse ranker in more detail.

**Training Sparse Ranker**: Let $\bar{\mathcal{Y}}^i = \{\bar{y}_j^i\}_{j=1}^{100}$ denote the set of top 100 predictions made by trained ELIAS model for training point $\mathbf{x}_i$. Similar to the representation used for clustering label space in

Table 6: Empirical prediction time, training time, and model sizes on benchmark datasets

| Dataset | Prediction (1 GPU) | Training (1 GPU) | Training (8 GPU) | Model Size |
|---|---|---|---|---|
| LF-AmazonTitles-131K | 0.08 ms/pt | 1.66 hrs | 0.33 hrs | 0.65 GB |
| Wikipedia-500K | 0.55 ms/pt | 33.3 hrs | 6.6 hrs | 2.0 GB |
| Amazon-670K | 0.57 ms/pt | 10.1 hrs | 2.1 hrs | 2.4 GB |
| Amazon-3M | 0.67 ms/pt | 37.6 hrs | 7.5 hrs | 5.9 GB |

stage 1 training, sparse ranker represents the input $\mathbf{x}^i$ with the static representation $\psi(\mathbf{x}^i)$ as:

$$\psi(\mathbf{x}^i) = [\frac{\mathtt{bow}(\mathbf{x}^i)}{\|\mathtt{bow}(\mathbf{x}^i)\|_2}, \frac{\phi(\mathbf{x}^i)}{\|\phi(\mathbf{x}^i)\|_2}] \tag{16}$$

It learns sparse linear classifiers $\bar{\mathbf{W}} = \{\bar{\mathbf{w}}_l\}_{l=1}^L$, where $\bar{\mathbf{w}}_l \in \mathbb{R}^{D'}$ and $D'$ is the dimensionality of $\psi$, on loss $\bar{\mathcal{L}}$ defined as following:

$$\bar{\mathcal{L}} = -\sum_{i=1}^N \sum_{l \in \bar{\mathcal{Y}}^i} (y_l^i \log(\sigma(\bar{\mathbf{w}}_l^T \psi(\mathbf{x}^i))) + (1 - y_l^i)(1 - \sigma(\bar{\mathbf{w}}_l^T \psi(\mathbf{x}^i)))) \tag{17}$$

Because these classifiers are only trained on $\mathcal{O}(100)$ labels per point, the complexity of $\bar{\mathcal{L}}$ is only $\mathcal{O}(100 \times N)$. Such sparse linear classifiers can be efficiently trained with second order parallel linear solvers like LIBLINEAR [12] on CPU. In particular, even on the largest Amazon-3M dataset with 3 million labels, training sparse ranker only takes about an hour on a standard CPU machine with 48 cores.

**Predicting with Sparse Ranker**: Similar to training, we first get top 100 predictions $\bar{\mathcal{Y}}^i$ from ELIAS model for each data point $\mathbf{x}^i$. Sparse classifiers are evaluated on each $(\mathbf{x}^i, l)$ pair where $l \in \bar{\mathcal{Y}}^i$. Let the score of ELIAS for the pair $(\mathbf{x}^i, l)$ be $p_l^i$ and score of sparse ranker be $q_l^i = \sigma(\bar{\mathbf{w}}_l^T \psi(\mathbf{x}^i))$. Ideally we would like the final score to be some combination of $p_l^i$ and $q_l^i$ but as observed in Section 3.5, these two scores are not very well calibrated across different label regimes. To correct this issue, we learn a score calibration module $\mathcal{T}$ which consists of a standard decision tree classifier[4] trained on a small validation set of 5000 data points. In particular, let the validation set be $\{(\mathbf{x}^i, \mathbf{y}^i)\}_{i=1}^{5000}$ and $\bar{\mathcal{Y}}^i$ denote the set of top 100 predictions made by ELIAS on validation point $\mathbf{x}^i$. Training data points for the score calibration module consists of all pairs $\bigcup_{i=1}^{5000} \bigcup_{l \in \bar{\mathcal{Y}}^i} (\mathbf{x}^i, l)$, where the input vector of a data point is a 4 dimensional vector $(p_l^i, q_l^i, p_l^i * q_l^i, f_l)$ and the target output is $y_l^i$. Here, $f_l$ denotes the training frequency (i.e. number of training points) of label $l$. During prediction, the final score for a pair $(\mathbf{x}^i, l)$ is returned as $\mathcal{T}(p_l^i, q_l^i, p_l^i * q_l^i, f_l) + p_l^i * q_l^i$.

### C.3 Practical Implementation and Resources Used

Many of the design choices for ELIAS's formulation is made to enable efficient implementation of the search index on GPU. For example, the row-wise sparsity constraint allows storing and operating the sparse adjacency matrix as two 2D tensors, which is much more efficient to work with on a GPU than a general sparse matrix. We implement the full ELIAS model excluding the sparse ranker component in PyTorch. Sparse ranker is implemented using LIBLINEAR utilities provided in PECOS[5] library. All experiments are run on a single A6000 GPU. Even on the largest dataset Amazon-3M with 3 million labels, prediction latency of single ELIAS model is about 1 ms per data point and training time is 50 hours.

### C.4 Additional Results

Table 7a reports the final accuracy numbers with different $\lambda$ on Amazon-670K dataset. With a very small $\lambda$ the loss only focuses on the classification objective which leads to significantly worse R@100 performance. Increasing $\lambda$ improves the overall performance up to a certain point, after that the performance saturates and starts degrading slowly. Table 7b reports the effect of choosing different $\kappa$

---

[4]https://scikit-learn.org/stable/modules/generated/sklearn.tree.DecisionTreeClassifier.html

[5]https://github.com/amzn/pecos

(row-wise sparsity parameter) to the final model performance on Amazon-670K dataset. We notice that the model performance increases up to a certain value of $\kappa$, after that the model performance (specially P@1) saturates and starts degrading slowly.

Table 7: ELIAS-1$^{(d)}$ results on Amazon-670K with (a) varying $\lambda$, (b) varying $\kappa$

(a)

| $\lambda$ | P@1 | P@5 | R@10 | R@100 |
|---|---|---|---|---|
| 0 | 47.80 | 39.45 | 49.17 | 66.05 |
| 0.01 | 48.30 | 39.86 | 49.73 | 67.78 |
| 0.02 | 48.48 | 39.94 | 49.96 | 68.27 |
| 0.05 | 48.68 | 40.05 | 50.33 | 68.95 |
| 0.1 | 48.72 | 40.05 | 50.19 | 68.91 |
| 0.2 | 48.62 | 39.96 | 50.06 | 68.82 |
| 0.5 | 48.48 | 39.76 | 49.80 | 68.55 |

(b)

| $\kappa$ | P@1 | P@5 | R@10 | R@100 |
|---|---|---|---|---|
| 100 | 46.79 | 36.60 | 42.90 | 56.38 |
| 200 | 47.88 | 38.67 | 46.96 | 63.30 |
| 500 | 48.68 | 40.04 | 49.99 | 68.48 |
| 1000 | 48.68 | 40.05 | 50.33 | 68.95 |
| 2000 | 48.58 | 40.07 | 50.27 | 68.91 |
| 5000 | 48.57 | 39.93 | 50.15 | 68.91 |
| 10000 | 48.32 | 39.73 | 49.97 | 68.84 |

Due to lack of space in the main paper, the full component ablation table is reported here in Table 8

| Method | P@1 | P@3 | P@5 | nDCG@3 | nDCG@5 | PSP@1 | PSP@3 | PSP@5 | R@10 | R@20 | R@100 |
|---|---|---|---|---|---|---|---|---|---|---|---|
| | | | | | LF-AmazonTitles-131K | | | | | | |
| Stage 1 | 36.96 | 24.67 | 17.69 | 37.47 | 39.21 | 28.29 | 33.16 | 37.44 | 47.69 | 51.74 | 58.81 |
| + Stage 2 | 37.90 | 25.61 | 18.45 | 38.83 | 40.76 | 29.73 | 35.16 | 39.88 | 50.12 | 54.62 | 62.88 |
| + Sparse ranker w/o calibration | 39.25 | 26.46 | 19.02 | 40.22 | 42.19 | 30.54 | 36.71 | 41.72 | 51.40 | 55.39 | 62.88 |
| + Score correction | 39.26 | 26.47 | 19.02 | 40.27 | 42.23 | 31.30 | 37.05 | 41.89 | 51.40 | 55.35 | 62.88 |
| + 3× ensemble | 40.13 | 27.11 | 19.54 | 41.26 | 43.35 | 31.05 | 37.57 | 42.88 | 53.31 | 57.79 | 65.15 |
| | | | | | Amazon-670K | | | | | | |
| Stage 1 | 46.63 | 41.65 | 37.58 | 44.02 | 42.11 | 29.89 | 33.20 | 35.66 | 46.08 | 52.29 | 61.72 |
| + Stage 2 | 48.68 | 43.78 | 40.04 | 46.24 | 44.68 | 31.22 | 34.94 | 38.31 | 50.33 | 57.67 | 68.95 |
| + Sparse ranker w/o calibration | 50.72 | 45.25 | 41.27 | 47.91 | 46.22 | 30.93 | 35.45 | 39.57 | 51.51 | 58.43 | 68.95 |
| + Score correction | 51.41 | 45.69 | 41.62 | 48.49 | 46.77 | 33.14 | 36.77 | 40.41 | 51.97 | 58.81 | 68.97 |
| + 3× ensemble | 53.02 | 47.18 | 42.97 | 50.11 | 48.37 | 34.32 | 38.12 | 41.93 | 53.99 | 61.33 | 72.07 |
| | | | | | Wiki-500K | | | | | | |
| Stage 1 | 76.54 | 57.65 | 44.33 | 69.54 | 67.01 | 32.61 | 40.04 | 43.48 | 65.78 | 72.06 | 80.60 |
| + Stage 2 | 77.81 | 59.14 | 45.85 | 71.22 | 68.97 | 33.38 | 41.88 | 45.98 | 68.33 | 74.97 | 84.70 |
| + Sparse ranker w/o calibration | 79.47 | 61.08 | 47.77 | 73.35 | 71.41 | 32.10 | 42.72 | 48.25 | 71.20 | 77.24 | 84.70 |
| + Score correction | 80.46 | 61.60 | 48.03 | 74.09 | 72.01 | 34.76 | 44.97 | 49.82 | 71.36 | 77.50 | 84.70 |
| + 3× ensemble | 81.26 | 62.51 | 48.82 | 75.12 | 73.10 | 35.02 | 45.94 | 51.13 | 72.74 | 79.17 | 87.22 |
| | | | | | Amazon-3M | | | | | | |
| Stage 1 | 49.12 | 46.31 | 44.10 | 47.46 | 46.26 | 16.32 | 19.44 | 21.57 | 19.12 | 27.90 | 49.15 |
| + Stage 2 | 49.93 | 47.07 | 44.85 | 48.20 | 46.97 | 14.97 | 17.46 | 19.34 | 18.94 | 28.28 | 52.93 |
| + Sparse ranker w/o calibration | 52.63 | 49.87 | 47.58 | 51.04 | 49.81 | 15.79 | 19.00 | 21.35 | 20.39 | 29.97 | 53.50 |
| + Score correction | 52.63 | 49.87 | 47.58 | 51.04 | 49.81 | 15.79 | 19.00 | 21.35 | 20.39 | 29.97 | 53.50 |
| + 3× ensemble | 54.28 | 51.40 | 49.09 | 52.65 | 51.46 | 15.85 | 19.07 | 21.52 | 21.59 | 31.76 | 57.09 |

Table 8: Full component ablation of ELIAS on all datasets

# D  Analysis of learned index

Table 9a reports the final accuracy numbers of ELIAS-1$^{(d)}$ model on Amazon-670K after threshold based pruning of the learned cluster-to-label assignments (i.e. for a particular threshold we remove all edges in the learned $\mathbf{A}$ which has smaller weight than the threshold). These results indicate that about $\sim 84\%$ edges can be pruned without hurting the model performance. Similarly, table 9b reports the final accuracy numbers of ELIAS-1$^{(d)}$ model on Amazon-670K after top-$K$ based pruning of the learned cluster-to-label assignments (i.e. we retain only top-$K$ label assignments per cluster).

Figure 7a plots the fraction of edges of the stage 1 tree that still remain in the learned adjacency matrix $\mathbf{A}$ after thresholding at various cutoff thresholds (i.e. for a threshold we only retain entries in which are greater than and evaluate how many edges of stage 1 tree remains). on Amazon-670K dataset. The plot reveals that almost $\sim 60\%$ stage 1 cluster assignments remain in the learned $\mathbf{A}$ with good confidence. Figure 7b plots the distribution of the average number of clusters assigned to a label for each label decile (decile 1 represents the head most decile and decile 10 represents the tail most decile). We say that a label $l$ is assigned to a cluster $c$ iff the weight $a_{c,l}$ in the learned adjacency

Table 9: ELIAS-1$^{(d)}$ results on Amazon-670K after pruning of learned cluster-to-label adjacency matrix $\mathbf{A}$ (a) after threshold based pruning (b) after top-k based pruning

(a)

| Threshold | % pruned | P@1 | P@5 | R@10 | R@100 |
|---|---|---|---|---|---|
| 0 | 0 | 48.68 | 40.04 | 50.33 | 68.95 |
| 0.01 | 20.89 | 48.68 | 40.05 | 50.33 | 68.96 |
| 0.05 | 64.42 | 48.68 | 40.04 | 50.33 | 68.96 |
| 0.1 | 73.63 | 48.68 | 40.04 | 50.33 | 68.95 |
| 0.25 | 84.52 | 48.65 | 40.02 | 50.26 | 68.82 |
| 0.5 | 89.11 | 48.40 | 39.48 | 48.98 | 66.75 |
| 0.75 | 91.95 | 47.70 | 38.19 | 46.38 | 62.17 |
| 0.9 | 93.13 | 47.26 | 37.42 | 44.91 | 59.53 |

(b)

| Top-$K$ | P@1 | P@5 | R@10 | R@100 |
|---|---|---|---|---|
| 1000 | 48.68 | 40.04 | 50.33 | 68.95 |
| 750 | 48.70 | 40.05 | 50.34 | 68.95 |
| 500 | 48.72 | 40.05 | 50.34 | 68.95 |
| 300 | 48.72 | 40.05 | 50.34 | 68.95 |
| 200 | 48.71 | 40.05 | 50.32 | 68.87 |
| 100 | 48.22 | 39.04 | 47.98 | 64.80 |
| 50 | 46.17 | 33.85 | 38.35 | 49.48 |

matrix $\mathbf{A}$ is greater than $0.25$. This demonstrates a clear trend that head labels get assigned to more number of clusters than tail labels.

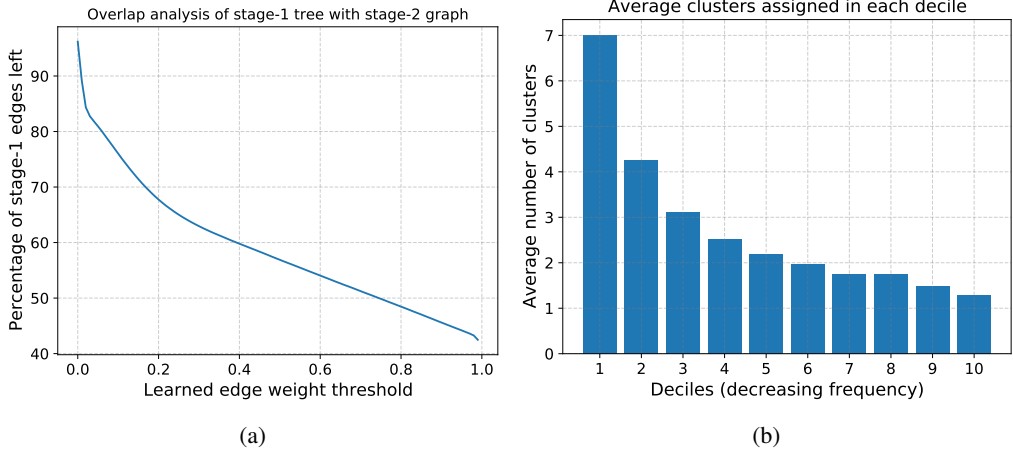

(a)                                    (b)

Figure 7: (a) percentage of stage 1 edges remaining in the learned adjacency matrix $\mathbf{A}$ at various cutoff thresholds on Amazon-670K dataset (b) decilewise distribution of the average number of assigned cluster in Amazon-670K dataset

In Figure 8 and 9, we qualitatively compare the training point distributions of labels which get assigned to multiple clusters and labels which get assigned to only one cluster by plotting TSNE plots of the training points of such labels and their assigned clusters. We say that a label $l$ is assigned to a cluster $c$ iff the weight $a_{c,l}$ in the learned adjacency matrix $\mathbf{A}$ is greater than $0.25$. These plots indicate that labels assigned to multiple clusters often have training points with a more multi-modal distribution than the labels which get assigned to only one cluster.

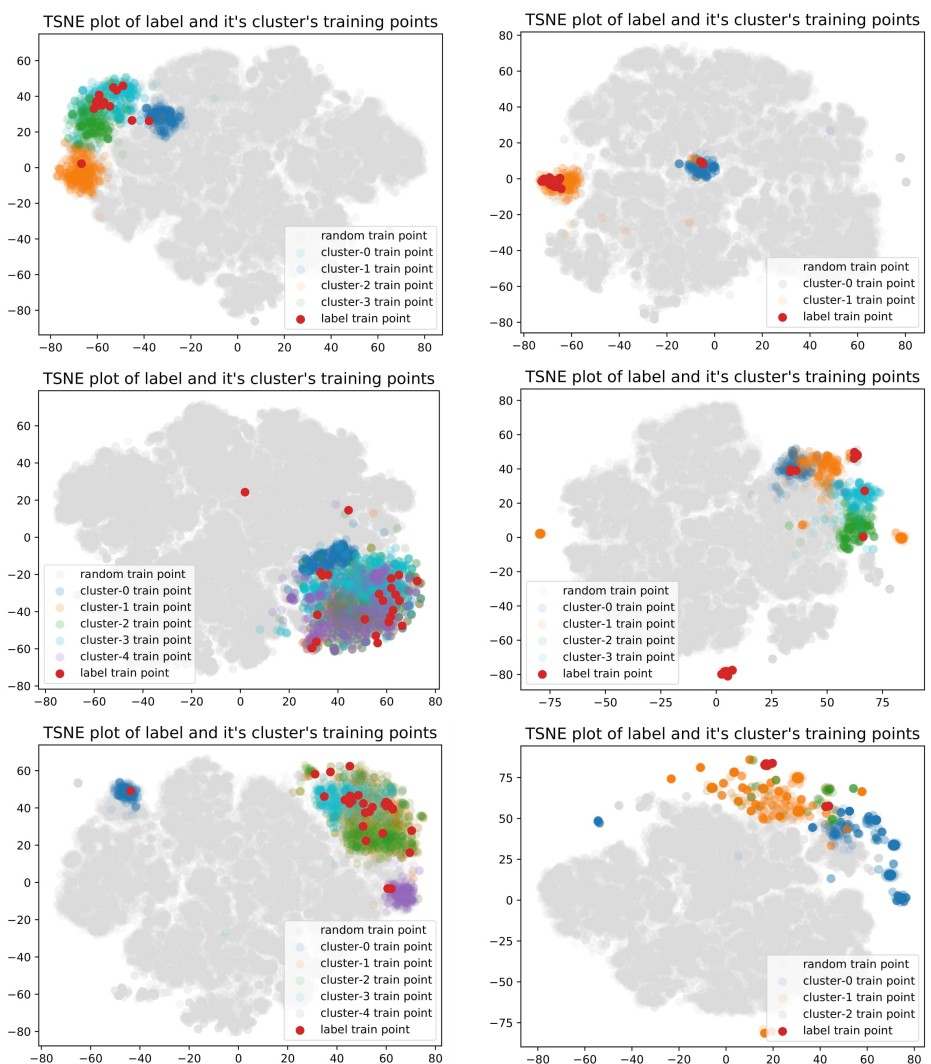

Figure 8: TSNE plot of training points of labels which get assigned to multiple cluster in the learned index structure on Amazon-670K dataset. We randomly sample 6 labels which have more than 1 but less than 6 edges with more than 0.25 learned weight ($a_{c,l}$). The red dots represent the training point of the sampled label and the dots in other colors indicate the training points of the respective assigned clusters (we say a training point $\mathbf{x}^i$ belongs to a cluster $c$ iff $s_c^i > 0.25$). As we can see training points of labels which gets assigned to multiple cluster often exhibit multi-modal distribution

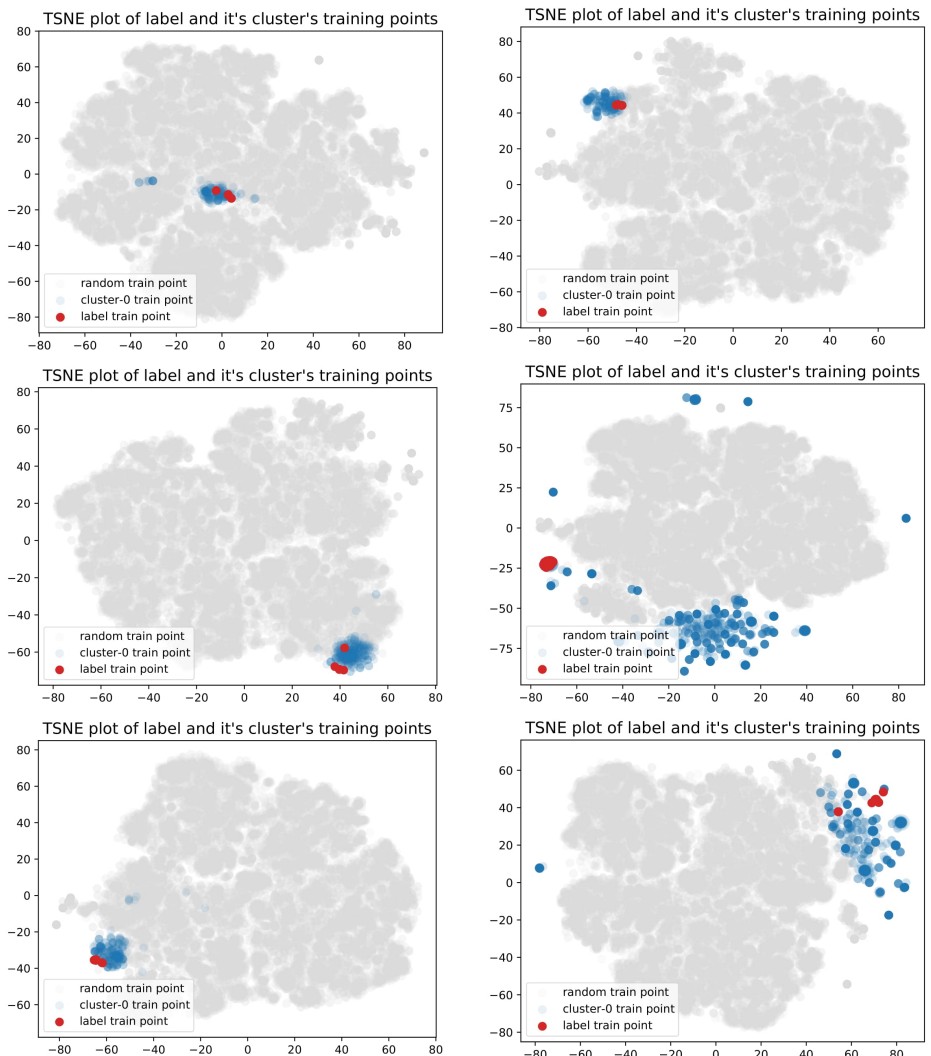

Figure 9: TSNE plot of training points of labels which get assigned to only one cluster in the learned index structure on Amazon-670K dataset. We randomly sample 6 labels which only have one edge with more than 0.25 learned weight ($a_{c,l}$). The red dots represent the training point of the sampled label and the blue dots indicate the training points of the assigned cluster (we say a training point $\mathbf{x}^i$ belongs to a cluster $c$ iff $s_c^i > 0.25$). As we can see training points of labels which gets assigned to only one cluster exhibit uni-modal distribution