# OpenReview forum: "ELIAS: End-to-End Learning to Index and Search in Large Output Spaces"
_NeurIPS.cc/2022/Conference — NeurIPS 2022 Accept_

### Official Review · Reviewer_i6NX · 2022-07-10

**Rating:** 7
**Confidence:** 4
**Soundness:** 4 excellent
**Presentation:** 4 excellent
**Contribution:** 4 excellent

**Summary:**

Tree-based methods are amongst the popular approaches for extreme multi-label learning problems. Existing approaches either use a decoupled strategy of training the label classifiers on top of a fixed tree index over labels, or alternate between updating the index and training the classifiers.

This paper proposes a relaxation of the tree-based index to a directed acyclic graph where each label can belong to multiple clusters with non-zero probability. The probability of label belonging to a cluster is learned together with label classifiers in an end-to-end fashion, and this leads to empirical improvement over using a fixed index over labels.

**Questions:**

Questions/Suggestions

Q1) Brute-force-OvA: Does this model consist of a BERT encoder followed by a multi-layer perceptron with L outputs? How much time did it take to train this model? Is it possible that BERT-OvA model also suffers from similar optimization challenges as training the proposed model from scratch? If so, then it is not necessarily a strong upper-bound on performance of these models.

Q2) What is the overall training time for the proposed model?

Q3) How were hyper-parameters such as $\kappa$ chosen? Was k-fold cross-validation used?

Q4) Why have previous XMC papers such as SiameseXML and DeepXML been not compared with?

Q5) The main set of results in the paper compare precision/recall of the proposed model with other baselines. But it might be interesting to have some results/insights/observations about the kind the structure of the index learnt using the proposed approach and how much it deviates from the fixed index from stage 1 training. Some ideas are:

 5a) In the final index structure, what fraction of labels get assigned to multiple clusters? Is it possible to prune assignment of a label to multiple clusters after training without affecting the accuracy of the model?

5b) In “Recall Comparison” para in Sec 4, authors hypothesize that improved performance for top 2 deciles is due to the tendency of popular labels to have multi-modal distribution. Is it the case that in the final index, popular labels tend to be assigned to multiple clusters and rare labels tend to pick a single cluster?

5c) How does row-sparsity affect the performance of the model?

Q6) Which dataset is used for Figure 5?

Q7) Instead of using a separate sparse ranker, why is the proposed model not trained with a combination of dense and sparse features for input as done in baseline methods such X-Transformer, Overlap-XMC?

**Limitations:**

Yes.

**Strengths And Weaknesses:**

Strengths

1) Proposed approach allows for learning label-to-cluster assignments in an end-to-end fashion which provides empirical improvements over using fixed label-to-cluster assignments.

2) The paper is well-written and easy to follow. The experiments support the main claim of the paper that jointly learning classifiers and index structure in an end-to-end fashion can yield improved performance over using a fixed index structure.

Weakness

1) Limited Analysis: It would be interesting to see analysis of the proposed approach beyond final accuracy numbers, for example, how much does the final index structure deviate from the fixed index structure used for stage 1 training.

2) Scalability: Since the proposed approach learns a two-level index, it is not immediately clear if the proposed approach can be applied to relax tree structures with three or more levels into a directed acyclic graph. This might limit the scalability of the proposed approach to larger industry scale datasets. However, this is a minor weakness as the approach is shown to scale well to and also outperform baselines on one of the largest publicly available extreme multi-label classification datasets (i.e. Amazon-3M).

---

> ### Author Response · Authors · 2022-08-02
> **Response to Reviewer i6NX**
>
> We thank reviewer i6NX for their very thorough feedback and helpful suggestions for understanding the learned index structure! We provide below additional analysis and answers to address the important questions raised by the reviewer.
>
> > **Analysis of the learned index**
>
> **Distribution of number of assigned clusters**:  Following the suggestion 5a we report the fraction of labels which get assigned to different number of clusters in the learned index. We say that a label $l$ is assigned to a cluster $c$ iff the weight $a_{c,l}$ in the learned adjacency matrix $\mathbf{A}$ is greater than $0.25$. Most labels ($\sim 80$%) get assigned to $\le 2$ clusters.
>
> *Table 12. Number of assigned clusters in learned index vs fraction of such labels in Amazon-670K dataset*
> | Num assigned clusters | 0 | 1 | 2 | 3-5 | 6-10 | 10+ |
> | -- | -- | -- | -- | -- | -- | -- |
> | Percentage of labels | 0.48% | 51.61% | 29.60% | 16.31% | 1.72% | 0.25% |
>
> **Overlap with the stage 1 fixed tree**: [here](https://www.dropbox.com/s/sxrr78b7t4w6etp/ELIAS_stage-1_vs_stage-2_overlap.pdf?dl=0) we plot the fraction of edges of the stage 1 tree that still remain in the learned adjacency matrix $\mathbf{A}$ after thresholding $\mathbf{A}$ at various cutoff thresholds (i.e. for a threshold $\gamma \in [0, 1]$ we only retain entries in $\mathbf{A}$ which are greater than $\gamma$ and evaluate how many edges of stage 1 tree remains). The plot reveals that almost $\sim 60$% stage 1 cluster assignments remain in the learned $\mathbf{A}$ with good confidence.
>
> **Threshold based pruning ablation**: in the following table we report the final accuracy numbers of ELIAS-1 model after cutoff threshold based pruning of the learned label-to-cluster assignments. These results indicate that about $\sim 84$% edges can be pruned without hurting the model performance.
>
> *Table 13. Accuracy numbers after threshold based pruning of learned label-to-cluster assignments on Amazon-670K dataset*
> | Cutoff Threshold | % of edges pruned | P@1 | P@5 | R@10 | R@100 |
> | --- | --- | --- | --- | --- | --- |
> | 0 | 0% | 48.68 | 40.04 | 50.33 | 68.95 |
> | 0.01 | 20.89% | 48.68 | 40.05 | 50.33 | 68.96 |
> | 0.05 | 64.42% | 48.68 | 40.04 | 50.33 | 68.96 |
> | 0.1 | 73.63% | 48.68 | 40.04 | 50.33 | 68.95 |
> | 0.25 | 84.52% | 48.65 | 40.02 | 50.26 | 68.82 |
> | 0.5 | 89.11% | 48.40 | 39.48 | 48.98 | 66.75 |
> | 0.75 | 91.95% | 47.70 | 38.19 | 46.38 | 62.17 |
> | 0.9 | 93.13% | 47.26 | 37.42 | 44.91 | 59.53 |
>
> **Top-k based pruning ablation**: in the following table we report the final accuracy numbers of ELIAS-1 model after top-k based pruning of the learned label-to-cluster assignments (i.e. we retain only top-k label assignments per cluster).
>
> *Table 14. Accuracy numbers after top-k based pruning of learned label-to-cluster assignments on Amazon-670K dataset*
> | Top-K | P@1 | P@5 | R@10 | R@100 |
> | --- | --- | --- | --- | --- |
> | 1000 | 48.68 | 40.04 | 50.33 | 68.95 |
> | 750 | 48.70 | 40.05 | 50.34 | 68.95 |
> | 500 | 48.72 | 40.05 | 50.34 | 68.95 |
> | 300 | 48.72 | 40.05 | 50.34 | 68.95 |
> | 200 | 48.71 | 40.05 | 50.32 | 68.87 |
> | 100 | 48.22 | 39.04 | 47.98 | 64.80 |
> | 50 | 46.17 | 33.85 | 38.35 | 49.48 |
>
> **$\kappa$ ablation**: in the following table we report the effect of choosing different $\kappa$ (row-wise sparsity parameter) to the final model performance on Amazon-670K dataset. We notice that the model performance increases up to a certain value of $\kappa$, after that the model performance (specially P@1) saturates and starts degrading slowly.
>
> *Table 15. $\kappa$ ablation on Amazon-670K*
> | $\kappa$ | P@1 | P@5 | R@10 | R@100 |
> | --- | --- | --- | --- | --- |
> | 100 | 46.79 | 36.60 | 42.90 | 56.38 |
> | 200 | 47.88 | 38.67 | 46.96 | 63.30 |
> | 500 | 48.68 | 40.04 | 49.99 | 68.48 |
> | 1000 | 48.68 | 40.05 | 50.33 | 68.95 |
> | 2000 | 48.58 | 40.07 | 50.27 | 68.91 |
> | 5000 | 48.57 | 39.93 | 50.15 | 68.91 |
> | 10000 | 48.32 | 39.73 | 49.97 | 68.84 |
>
> **Effectiveness on multi-modal labels**: Please refer to our response to reviewer ubcH for a quantitative and qualitative analysis of learned index's behaviour on labels which get assigned to multiple clusters.
>
> > **Scalability**
>
> With a reduced embedding dimension ELIAS can scale to datasets with up to 10M labels even on a single GPU but we do agree that scaling ELIAS to much bigger datasets (100M or 1B scale) will require extending the two-level index to deeper hierarchies and is one of the future directions we hope to explore.

---

> > ### Author Response · Authors · 2022-08-02
> > **Response to Reviewer i6NX - part 2**
> >
> > > **Brute-force OvA details**
> >
> > Yes, Bert-OvA baseline is BERT encoder followed by a linear classification layer with L outputs. It is unlikely that this approach suffers from the same optimization challenges as ELIAS since it doesn’t have any moving assignments i.e. the training feedback that the model gets is always consistent because we always know what are the right labels for a given training point. In ELIAS the major challenge is that since there’s no unique path from the root to a particular label $l$, we don’t have this explicit training signal that what are the right clusters for a training point, this leads to the optimization challenge when jointly training every component of the model from random initialization. Note that this doesn’t happen when training with a fixed index structure where a label is uniquely assigned to a cluster because if a label is uniquely assigned then the right clusters for a training point are always going to be clusters of the positive labels.
> >
> > > **What is the overall training time for the proposed model**
> >
> > Please refer to Table 8 in our response to reviewer BfTa
> >
> > > **How were hyper-parameters such as  $\kappa$ chosen? Was k-fold cross-validation used?**
> >
> > Most of the hyperparameters such as $\kappa$, $\lambda$, etc are tuned only on the smallest LF-AmazonTitles-131K dataset, on the rest of the bigger datasets we only tune learning rate on a small held-out validation set
> >
> > > **Why have previous XMC papers such as SiameseXML and DeepXML been not compared with?**
> >
> > DeepXML numbers are reported in Table 1 under the name of Astec since the DeepXML paper refers "DeepXML" name as the framework, and the method as "Astec". We don’t compare with SiameseXML because it uses additional label features which most of the standard XMC methods don’t use nor do the standard XMC datasets have these label features (Amazon-670K, Wikipedia-500K, Amazon-3M).
> >
> > > **Which dataset is used for Figure 5?**
> >
> > Amazon-670K is used for Figure 5, we'll update the figure caption to mention this
> >
> > > **Instead of using a separate sparse ranker, why is the proposed model not trained with a combination of dense and sparse features for input as done in baseline methods such X-Transformer, Overlap-XMC?**
> >
> > Joint training is not possible when learning on combination of dense and sparse features because currently, no deep learning frameworks (pytorch, tf, etc) support efficient learning with sparse features. X-Transformer, Overlap-XMC decouple learning of the deep encoder from the learning of the classifiers i.e. they first learn their deep encoder on the matching task with only dense features, they then obtain dense representations from the encoder and learn the ranker classifiers level by level on the concatenated fixed dense and sparse representations of the input using convex LIBLINEAR solvers.
> >
> > ---
> > We hope that our response helped in addressing your concerns, we'll be happy to answer/discuss any further clarifications needed.

---

> > > ### Comment · Reviewer_i6NX · 2022-08-04
> > > **Reply to author response**
> > >
> > > Thank you for running these additional experiments and analysis! I hope that you will be able to add this analysis to the main paper or to the appendix!
> > > I have a few minor lingering questions
> > > 1) It is still not clear to me how the validation data was created. Were labels for validation data sampled uniformly at random? If so, then validation data is more likely to have head labels (i.e. labels with high frequency). Also, it might be possible that a rare label which occurred only once in training data gets moved to validation split. Since the proposed approach can not handle new labels at test time, it will fail for such rare labels that get moved to validation split. I would appreciate some more clarification on this.
> > > 2) How long did it take to train Brute-force OvA baseline?
> > > 3) In order to better understand the contribution of sparse re-ranker towards overall performance, it might be helpful to breakdown performance of ELIAS and ELIAS++ based on label frequency. If ELIAS++ consistently improves over ELIAS across all label frequency buckets, then it might suggest that the reason behind ELIAS++ being better than ELIAS is simply because of deep encoder used in ELIAS receiving truncated text while ELIAS++ used the entire text (by converting them to sparse features).
> > > Although clearly beyond the scope of this paper, it might be useful to investigate models such as Longformer [1] and Big-Bird[2] which can encode long documents more efficiently than BERT, thus potentially avoiding the need of any sparse re-ranking model.
> > >
> > > [1] Beltagy, Iz, Matthew E. Peters, and Arman Cohan. "Longformer: The long-document transformer." arXiv preprint arXiv:2004.05150 (2020).
> > >
> > > [2] Zaheer, Manzil, et al. "Big bird: Transformers for longer sequences." Advances in Neural Information Processing Systems 33 (2020): 17283-17297.

---

> > > > ### Author Response · Authors · 2022-08-05
> > > > **Response to reviewer's followup questions**
> > > >
> > > > Yes, we do plan to put results from all the additional experiments and a summary of the rebuttal discussion in the main paper/appendix.
> > > >
> > > > (1) Following other methods like AttentionXML and LightXML, we create the validation set by randomly sampling 5000 points from the training set, this is usually $\le$ 1% of the training data for most of the large extreme classification datasets (Amazon-670K, Wikipedia-500K, Amazon-3M). You're correct that the validation set is more likely to have head labels but so is the whole extreme classification dataset since the random sample represents a smaller unbiased version of the full dataset. Yes, that's definitely a possibility that the validation split completely removes extremely tail labels with 1 or 2 data points from the training set but usually, these labels don't contribute to the final prediction accuracy anyway since label classifiers learned on just 1 or 2 training points is often not of good quality and tends to overfit to those particular training points. One way to overcome the problem of information loss in the validation set is to re-learn the model on the full training set after we have validated the hyperparameters on the split training and validation set. In practice, this gives very minor improvements for the extra computation cost it incurs.
> > > >
> > > > (2) The Bert-OvA model took $\sim$18 hours to train on Amazon-670K dataset
> > > >
> > > > (3) Table 16 below reports the contribution to P@5 of each decile on Amazon-670K dataset. Sparse re-ranker improves performance in almost all deciles. As you mentioned, we believe a major factor could be the truncated text used in deep encoder but there are other factors which might favor the addition of sparse re-ranker for e.g. a) sparse classifiers have a much bigger parameter space than dense classifiers, b) since sparse re-ranker only ranks the top 100 labels generated by ELIAS, it's addition acts in a similar way as boosting where sparse re-ranker's job is to correct some of the mistakes made in the top 100. Thanks for the suggestions on using Longformer and Big-Bird, we agree with you that there is a need to investigate efficient encoders for large-scale classification settings because even with the truncated text of length 128 the computational cost of BERT models ($T_{\text{bert}}$) is significantly high which makes them hard to use in practical settings and becomes one of the major computational bottleneck in any state of the art method.
> > > >
> > > > *Table 16. ELIAS-1 vs ELIAS-1 + sparse re-ranker decilewise contribution to P@5 on Amazon-670K dataset*
> > > > |  Method | 1 | 2 | 3 | 4 | 5 | 6 | 7 | 8 | 9 | 10 | P@5 |
> > > > | --- | --- | --- | --- | --- | --- | --- | --- | --- | --- | --- | --- |
> > > > | ELIAS-1 | 7.35 | 6.57 | 5.44 | 4.48 | 3.73 | 3.10 | 2.59 | 2.33 | 1.74 | 2.72 | 40.04 |
> > > > | ELIAS-1 + sparse re-ranker | 7.75 | 6.80 | 5.52 | 4.51 | 3.70 | 3.05 | 2.60 | 2.35 | 1.82 | 3.16 | 41.27 |
> > > >
> > > > We're happy to discuss any further questions/clarifications needed

---

> > > > > ### Comment · Reviewer_i6NX · 2022-08-05
> > > > > **Some more follow-up questions**
> > > > >
> > > > > Thank you for the clarification!! This is really great work!! I updated my review rating!
> > > > >
> > > > > Re: Brute-force OvA baseline:
> > > > > So you trained a separate classifier for every label, right? How did you mine negative examples for a label? Did you use all datapoints $x_i \in X_{train}$ such that $y_{i, \ell} = 0$ as negatives when training classifier for label $\ell$ or did you do some form of subsampling?
> > > > >
> > > > > Is the following a correct description of the classifier?
> > > > > First encode an instance $x$ using BERT and pass the encoded representation through a linear classifier corresponding to label $\ell$ to classify $x$ as 0/1 wrt label $\ell$.
> > > > >
> > > > > Did you train OvA classifiers by stacking a linear layer with $|\mathcal{L}|$-dim output on top of BERT encoder and using sigmoid over each output unit?
> > > > >
> > > > > Are BERT model parameters frozen or trained together with linear classifier parameters? If they are trained together, then I am a little surprised that it was possible to train the model in 18 hours! (unless there is some sort of subsampling involved or BERT model parameters are frozen).
> > > > > Also, what does L323 mean when you say "We follow the same training procedures as ELIAS for this baseline"?
> > > > >
> > > > >
> > > > > Re: Using Dense + Sparse Features
> > > > >
> > > > > I understand how it might not be possible to perform end-to-end learning with dense+sparse features using existing deep learning frameworks. I was reading through the supplement and realized that dense+sparse features were used for clustering **before** stage 1 training.
> > > > > How were the dense features obtained here? Did you use some pretrained BERT model or did you use a trained model from previous papers such X-Transformer?
> > > > >
> > > > > Just curious (no need to run this experiment for the rebuttal), instead of training a separate re-ranker using dense+sparse features, would it be reasonable to train a model with sparse+dense features on the final index obtained while keeping dense model fixed and the index fixed? Or will there be some complications due to labels belonging to multiple clusters?

---

> > > > > > ### Author Response · Authors · 2022-08-06
> > > > > > **Response**
> > > > > >
> > > > > > Many thanks for your appreciative comments and for upgrading your score! Please find below our response to the follow-up questions
> > > > > >
> > > > > > > So you trained a separate classifier for every label, right?
> > > > > >
> > > > > > Yes
> > > > > >
> > > > > > > How did you mine negative examples for a label? Did you use all datapoints $x_i \in X_{train}$ such that $y_{i, l} = 0$ as negatives when training classifier for label $\ell$ or did you do some form of subsampling?
> > > > > >
> > > > > > You're correct, all training points which are not positive get counted as the negative, there is no subsampling performed.
> > > > > >
> > > > > > > Is the following a correct description of the classifier? First encode an instance $x$ using BERT and pass the encoded representation through a linear classifier corresponding to label $\ell$ to classify as 0/1 wrt label $\ell$. Did you train OvA classifiers by stacking a linear layer with $\lvert \mathcal{L} \rvert$-dim output on top of BERT encoder and using sigmoid over each output unit?
> > > > > >
> > > > > > That's correct, the whole Bert-OvA model is BERT encoder followed by a $d \times \lvert \mathcal{L} \rvert$ classifier matrix where each column represents a label $\ell \in \mathcal{L}$. Forward pass of this model will generate $\lvert \mathcal{L} \rvert$ outputs which are passed through sigmoid function to produce logits.
> > > > > >
> > > > > > > Are BERT model parameters frozen or trained together with linear classifier parameters? If they are trained together, then I am a little surprised that it was possible to train the model in 18 hours! (unless there is some sort of subsampling involved or BERT model parameters are frozen)
> > > > > >
> > > > > > BERT parameters are trained together with the linear classifier parameters. Basic operations like matrix multiplication are very heavily parallelized on GPU implementations, because of this even brute-force matrix multiplication on a label space of 670K labels is manageable.  On moderately sized datasets ($\sim$500K), the computational cost of the BERT encoder makes it hard to observe a stark difference in the training times between the brute-force baseline and any subsampling-based XMC method (which uses BERT). Although, as the label space grows ($>$1M) the difference grows larger and larger because then the cost of brute-force matrix multiplication starts to heavily outweigh the constant cost of the BERT encoder.
> > > > > >
> > > > > > > Also, what does L323 mean when you say "We follow the same training procedures as ELIAS for this baseline"?
> > > > > >
> > > > > > We meant to say that we keep the same training setup as ELIAS's code (i.e. AdamW optimizer, mixed precision training, etc) for implementing this baseline
> > > > > >
> > > > > > > ... How were the dense features obtained here? Did you use some pretrained BERT model or did you use a trained model from previous papers such X-Transformer?
> > > > > >
> > > > > > We experimented with two options here 1) obtain clusters using pre-trained BERT and train stage 1 model, 2) obtain clusters using pre-trained BERT and train stage 1 model for a few epochs, then recompute clusters based on current BERT embeddings and continue stage 1 training for the remaining epochs. On Wikipedia-500K and Amazon-3M, we don't observe any significant difference in final accuracy, on Amazon-670K the second approach gives slightly better results.
> > > > > >
> > > > > > > Would it be reasonable to train a model with sparse+dense features on the final index obtained while keeping dense model fixed and the index fixed? Or will there be some complications due to labels belonging to multiple clusters?
> > > > > >
> > > > > > We believe it should be possible to train such a model and in some sense, ELIAS's sparse re-ranker is trying to do something similar but it's training only the leaf layer. Training each layer one at a time seems reasonable although how to assign training points to cluster nodes is not very straightforward.

---

### Official Review · Reviewer_ubcH · 2022-07-11

**Rating:** 6
**Confidence:** 4
**Soundness:** 2 fair
**Presentation:** 3 good
**Contribution:** 3 good

**Summary:**

The paper addresses a relevant problem in partition based approach to extreme multi-label classification (XMC), where existing methods have 2 stages – the first stage uses a shallow tree-based index to hard partition the label space and the second stage learns classifiers into labels inside that partition. The challenge is that if all the labels are not inside the partition in the first stage, then they cannot be in the output of the second stage at all. This paper introduces a variant, ELIAS, where the first stage is replaced by a weighted graph based index with soft learnable parameters that are learned together with the final task classification objective. Experimental results on popular XMC datasets show reasonable improvements in precision/recall metrics over existing XMC methods.



**Questions:**

The authors claim earlier in section 1 that ELIAS is better suited to handled multi-modal label distributions. However, this aspect is not really demonstrated in the paper. They hypothesize in the recall comparison section in section 4 that the increased recall might be due to the  multi-modal distribution of popular labels. Do they have any quantitative evidence to back this claim?

**Ethics Review Area:**

["I don’t know"]

**Limitations:**

This paper addresses a technical improvement in XMC. As such, I don’t see any significant potential negative societal impact of this specific work.

**Strengths And Weaknesses:**

Originality:
The general idea of training the retrieval stage using an objective (supervised, unsupervised or self-supervised) followed by a classification stage is becoming more commonplace now, with works like REALM for language modeling, and retrieval augmented convolutional networks in computer vision. However, this paper appears to be the first work applying jointly training the retrieval and classification stages in XMC for multi-label classification. The authors have taken a natural next step in evolving this general idea.
Quality:
The paper takes a reasonable approach – motivating the problem, challenges with existing approaches to XMC, proposing a solution, describing the details and testing it on datasets that show the value of this approach.
Clarity:
The paper is reasonably well written and easy to understand, with sufficient details. While I have not looked at the code, I am glad that they have shared the PyTorch implementation for transparency and enable reproducibility.
Significance:
XMC is an important problem in a number of domains where the output space of labels is large – many information retrieval problems including search, ads, commerce etc. are in this category. As such the work addresses a significant problem in XMC.

---

> ### Author Response · Authors · 2022-08-02
> **Response to reviewer ubcH**
>
> We thank reviewer ubcH for their valuable comments and appreciate their understanding of the contribution of this work in the context of recent advances in deep learning and XMC! We provide below additional discussion to make a stronger argument for the claim that ELIAS is better suited for multi-modal label distributions.
>
> **Qualitative analysis**: We qualitatively compare the training point distributions of labels which get assigned to multiple clusters and labels which get assigned to only one cluster by plotting TSNE plots of the training points of such labels and their assigned clusters [here](https://www.dropbox.com/s/eqfw06dk66quo3r/ELIAS_visualize_multimodal_label_training_points.pdf?dl=0). We say that a label $l$ is assigned to a cluster $c$ iff the weight $a_{c,l}$ in the learned adjacency matrix $\mathbf{A}$ is greater than $0.25$. These plots indicate that labels assigned to multiple clusters often have training points with a more multi-modal distribution than the labels which get assigned to only one cluster.
>
> **XR-Transformer vs ELIAS comparison**: in the following table we compare the contribution to R@100 of labels belonging to different label bins for XR-Transformer-1 and ELIAS-1. Here label bins are created based on number of assigned clusters in learned ELIAS model (for e.g. column 2 presents the contribution to R@100 of all labels which have only 1 clusters assigned to them). The results indicate that the relative improvement in performance in the ELIAS model over XR-Transformer is much more significant for labels which get assigned to multiple clusters than labels which only get assigned to single cluster.
>
> *Table 10. R@100 contribution of labels with different numbers of clusters assigned to them in Amazon-670K dataset*
> Num assigned clusters | 1 | 2 | 3-5 | 6-10 | 10+
> |--|--|--|--|--|--|
> | R@100 contribution (XR-Transformer-1) | 24.26 | 16.91 | 17.19 | 4.72 | 1.56
> | R@100 contribution (ELIAS-1) | 25.10 | 17.70 | 18.59 | 5.47 | 2.06
> | Delta | +3.4% | +4.6% | +8.1% | +15.8% | +32.0%
>
> **Decilewise distribution of number of assigned clusters**: in the following table we analyze the distribution of the average number of clusters assigned to a label for each label decile (decile 1 represents the head most decile and decile 10 represents the tail most decile). This demonstrates a clear trend that head labels get assigned to more number of clusters than tail labels in the learned assignments.
>
> *Table 11. Decilewise distribution of the average number of assigned cluster in Amazon-670K dataset*
> | Deciles | 1 | 2 | 3 | 4 | 5 | 6 | 7 | 8 | 9 | 10 |
> |--|--|--|--|--|--|--|--|--|--|--|
> | **Avg. assigned clusters** | 6.99 | 4.25 | 3.10 | 2.50 | 2.19 | 1.96 | 1.74 | 1.74 | 1.49 | 1.29
>
> ---
>
> We hope that our response helped in addressing your concerns, we'll be happy to answer/discuss any further clarifications needed.

---

> ### Comment · Area_Chair_GYs6 · 2022-08-09
> **Author rebuttal phase closing**
>
> The author-rebuttal phase closes today. Please acknowledge the author rebuttal and state if your position has changed. Thanks!

---

### Official Review · Reviewer_BfTa · 2022-07-11

**Rating:** 7
**Confidence:** 3
**Soundness:** 3 good
**Presentation:** 3 good
**Contribution:** 3 good

**Summary:**

This paper focuses on the extreme multi-label classification (XMC) problem and proposes a method called ELIAS.
(1) ELIAS adopts a two-layer index for representing extreme-scale labels, and it learns overlapping cluster partitions by assigning each label to multiple clusters.
(2) ELIAS adopts a two-staged training strategy, where a XMC model is trained with fixed cluster partition generated by k-balanced clustering in the first stage, the cluster partition is generated according to the weighted count of labels assigning to corresponding clusters according to the XMC model trained in first stage, and the XMC model with value of the cluster partition are trained in the second stage.
(3) ELIAS++ adopts a sparse ranker and a calibration module for achieving further improvement compared to ELIAS.
Experiments are conducted on benchmark dataset, where ELIAS achieves better performance compared to previous SOTA methods.

**Questions:**

How does the loss (i.e., classification and shortlist loss) affect the performance of ELIAS? Is it possible to add an ablation study of $\lambda$?


**Ethics Review Area:**

["I don’t know"]

**Strengths And Weaknesses:**

Strength:
+ This paper is well motivated and the proposed method achieves significant performance compared to previous SOTA methods.
+ The writing is good and easy to follow.

Weakness:
- It is not rigorous to say that ELIAS is an end-to-end method, though Eq. 7 and Eq. 8 can be optimized in an end-to-end manner in theory. In practice, the model is trained in a two-stage manner, and the label-to-cluster assignment is generated according to Eq. 10, which is non-differentiable.
- Experiment results of ELIAS^{(d)} in Table 1 have not be analyzed in Table 3. Is ELIAS^{(d)} corresponds to Stage 1+Stage 2+3 x ensemble in Table 3?
- Lack of time complexity analysis. It will be better to add a discussion about empirical wall-clock time of the two-stage training process of ELIAS.

---

> ### Author Response · Authors · 2022-08-02
> **Response to Reviewer BfTa**
>
> We would like to thank reviewer BfTa for their valuable and constructive feedback! Below we provide additional discussion and answers to address the valid concerns raised by the reviewer.
>
> **Time complexity analysis**: the time complexity for processing a batch of $\eta$ data-points is $\mathcal{O}(\eta(T_{\text{bert}} + Cd + b\kappa + Kd))$ where $T_{\text{bert}}$ represents the time complexity of the bert encoder, $C$ represents the number of clusters in index, $d$ is the embedding dimension, $b$ is the beam size, $\kappa$ is the row-wise sparsity of label-to-cluster adjacency matrix $A$, and $K$ is the number of labels shortlisted for classifier evaluation. Assuming $C = \mathcal{O}(\sqrt{L})$,  $\kappa = \mathcal{O}(L/C) = \mathcal{O}(\sqrt{L})$ and $K = \mathcal{O}(\sqrt{L})$, the final time complexity comes out to be $\mathcal{O}(\eta(T_{\text{bert}} + \sqrt{L}(2d + b)))$.
>
> In practice, because of ELIAS’s shallow index and design choices such as a $\kappa$-row-wise sparse adjacency matrix, each computation involved in the forward pass can be written as tensor operations which are highly parallelizable on a GPU. This results in very fast inference times when doing the inference on GPUs.
>
> **Empirical runtimes**: the following table provides the empirical runtimes and model sizes for the ELIAS-1$^{(d)}$ model. All reported numbers are for A6000 GPU with 24 core standard CPU machine. The reported training times are the total training times i.e. both stage 1 and stage 2 training. Prediction times are reported as average prediction time per point when doing batch prediction.
>
> *Table 8. Empirical prediction time, training time, and model sizes on benchmark datasets*
> |  Dataset | Prediction (1 GPU) | Training (1 GPU) | Training (8 GPU) | Model Size |
> | --- | --- | --- | --- | --- |
> | **LF-AmazonTitles-131K** | 0.08 ms/pt | 1.66 hrs | 0.33 hrs | 0.65 GB |
> | **Wikipedia-500K** | 0.55 ms/pt | 33.3 hrs | 6.6 hrs | 2.0 GB |
> | **Amazon-670K** | 0.57 ms/pt | 10.1 hrs | 2.1 hrs | 2.4 GB |
> | **Amazon-3M** |  0.67 ms/pt | 37.6 hrs | 7.5 hrs | 5.9 GB |
>
> **$\lambda$ ablation**: in the following table we report the final accuracy numbers with different $\lambda$ on Amazon-670K dataset. With a very small $\lambda$ the loss only focuses on the classification objective which leads to significantly worse R@100 performance, increasing $\lambda$ improves the overall performance up to a certain point, after that the performance saturates and starts degrading slowly.
>
> *Table 9. Final accuracy numbers on Amazon-670K with varying $\lambda$*
> | $\lambda$ | P@1 | P@5 | R@10 | R@100 |
> | --- | --- | --- | --- | --- |
> | **0** | 47.80 | 39.45 | 49.17 | 66.05 |
> | **0.01** | 48.30 | 39.86 | 49.73 | 67.78 |
> | **0.02** | 48.48 | 39.94 | 49.96 | 68.27 |
> | **0.05** | 48.68 | 40.05 | 50.33 | 68.95 |
> | **0.1** | 48.72 | 40.05 | 50.19 | 68.91 |
> | **0.2** | 48.62 | 39.96 | 50.06 | 68.82 |
> | **0.5** | 48.48 | 39.76 | 49.80 | 68.55 |
>
> > **It is not rigorous to say that ELIAS is an end-to-end method...**
>
> We consider ELIAS to be end-to-end because the stage 2 training (which represents the main contribution of our work) jointly trains the representation, indexing, and classification parameters in an end-to-end fashion allowing each component to adapt to each other w.r.t. final objective. We do acknowledge that the current solution involves a careful initialization of model parameters (based on stage 1 training) before the end-to-end training begins which is not ideal.
>
> > **Experiment results of ELIAS^{(d)} in Table 1 have not been analyzed in Table 3...**
>
> ELIAS$^{(d)}$ in Table 1 is indeed Stage 1+Stage 2+3 x ensemble, thanks for pointing it out, we'll make this explicit in the main text.
>
> ---
>
> We hope that our response helped in addressing your concerns, we'll be happy to answer/discuss any further clarifications needed.

---

> > ### Comment · Reviewer_BfTa · 2022-08-09
> > **Reply to author's response**
> >
> > Thanks for the authors' response and it has addressed most of my concerns. I'll raise my score accordingly.

---

> > > ### Author Response · Authors · 2022-08-09
> > > **Thank you**
> > >
> > > We're happy to know that our response helped in addressing your concerns. Thanks again for the helpful feedback and upgrading the score!

---

### Meta-Review · Area_Chair_GYs6 · 2022-08-28

**Recommendation:** Accept
**Confidence:** Certain

**Metareview:**

The paper considers extreme multilabel classification (XMC) and proposes a two-stage retrieval and classification model which replaces the usual initial hard-partitioning with a soft learnable partitioning. The reviewers concur that the end-to-end methodology for jointly training the representation, indexing, classification parameters is novel and leads to notable improvements over the performance of current SOTA XMC methods.

**Award:**

No

---

### Decision · Program_Chairs · 2022-09-14

Accept